# Reasoning-Aware Multimodal Fusion for Hateful Video Detection

**Shuonan Yang**                                                      *sy446@exeter.ac.uk*
*Multimodal Intelligence Lab*
*Department of Computer Science*
*University of Exeter*

**Tailin Chen**                                                      *T.Chen2@exeter.ac.uk*
*Multimodal Intelligence Lab*
*Department of Computer Science*
*University of Exeter*

**Jiangbei Yue**                                                      *J.Yue@exeter.ac.uk*
*Multimodal Intelligence Lab*
*Department of Computer Science*
*University of Exeter*
*School of Computer Science*
*University of Leeds*

**Guangliang Cheng**                                      *Guangliang.Cheng@liverpool.ac.uk*
*School of Computer Science and Informatics*
*University of Liverpool*

**Jianbo Jiao**                                                      *J.Jiao@bham.ac.uk*
*Machine Intelligence + x Group*
*School of Computer Science*
*University of Birmingham*

**Zeyu Fu**[*]                                                      *Z.Fu@exeter.ac.uk*
*Multimodal Intelligence Lab*
*Department of Computer Science*
*University of Exeter*

**Reviewed on OpenReview:** *https://openreview.net/forum?id=U9KnNiuMu1*

## Abstract

Hate speech in online videos is posing an increasingly serious threat to digital platforms, especially as video content becomes increasingly multimodal and context-dependent. Existing methods often struggle to effectively fuse the complex semantic relationships between modalities and lack the ability to understand nuanced hateful content. To address these issues, we propose an innovative Reasoning-Aware Multimodal Fusion (RAMF) framework. To tackle the first challenge, we design Local-Global Context Fusion (LGCF) to capture both local salient cues and global temporal structures, and propose Semantic Cross Attention (SCA) to enable fine-grained multimodal semantic interaction. To tackle the second challenge, we introduce contrastive reasoning—a structured three-stage process where a vision-language model generates (i) objective descriptions, (ii) hate-assumed inferences, and (iii) non-hate-assumed inferences—providing complementary semantic perspectives that enrich the model's contextual understanding of nuanced hateful intent. Evaluations on two real-world hateful video datasets demonstrate that our method achieves robust generalisa-

---

[*]Corresponding author.

tion performance, improving upon state-of-the-art methods by 3% and 7% in Macro-F1 and hate class recall, respectively. The source codes and data required to reproduce our results are available at `https://github.com/Multimodal-Intelligence-Lab-MIL/RAMF`.

**Disclaimer: This paper contains sensitive content that may be disturbing to some readers.**

# 1 Introduction

Online videos have become a dominant communication medium, and their widespread reach has enabled hateful content to spread rapidly (Das et al., 2023). Such content exacerbates discrimination and social division, and can even incite offline violence (Townsend, 2025; Robertson, 2025). Current methods (Zhang et al., 2024; Koushik et al., 2025; Wang et al., 2024a; Das et al., 2023; Yue et al., 2025; Sun et al., 2026) follow a standard of extracting and fusing features from video frames, audio, and transcribed text, but lack effective multimodal semantic interaction. As illustrated in Figure 1, existing hateful video detection faces two key challenges: Challenge 1: Nuanced Context Understanding. Hateful videos often convey harmful intent through nuanced contextual cues spanning time and modalities, such as the temporal resonance between specific visuals and statements (Yang et al., 2025; Wang et al., 2025b), or the implicit linkage between visual focus and auditory content (Rehman et al., 2025). Challenge 2: Multimodal Semantic Fusion. Current methods (Zhang et al., 2024; Koushik et al., 2025; Wang et al., 2024a; Das et al., 2023; Yue et al., 2025) follow a standard of extracting and fusing features from video frames, audio, and transcribed text, but lack effective multimodal semantic interaction. Both challenges demand deep contextual reasoning capabilities that standard pipelines struggle to provide.

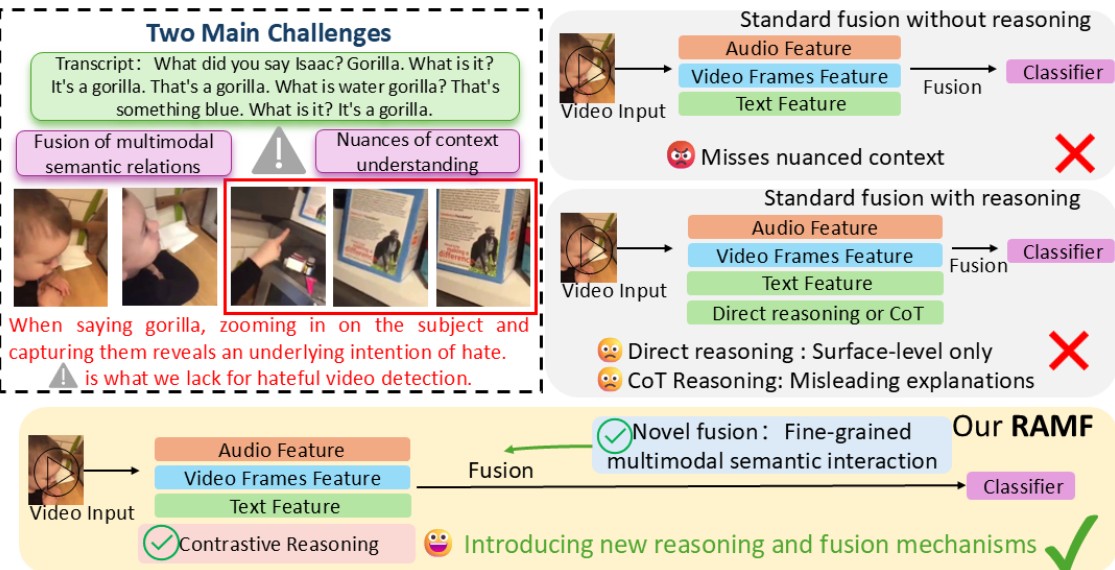

Figure 1: Left: Two main challenges—fusion of multimodal semantic relations and nuances of context understanding. Right: Standard paradigms vs. our RAMF.

Recent approaches attempt to enhance contextual understanding through visual-language models (VLMs) generated reasoning (Lang et al., 2025; Hee et al., 2024). As shown in Figure 1, this standard fusion with reasoning methods incorporates direct reasoning or Chain-of-Thought (CoT) reasoning (Hee et al., 2025) as additional inputs. However, direct reasoning produces only surface-level descriptions lacking hateful semantic associations, while recent work reveals that CoT may generate misleading explanations that fail to reflect actual model reasoning (Barez et al., 2025), raising reliability concerns for sensitive detection tasks. Beyond reasoning quality, recent research (Wang et al., 2025b; Yang et al., 2025) of the hateful video dataset reveals that hate cues exhibit heterogeneous temporal distributions: they may erupt briefly within short segments

or be dispersed across the entire video timeline. Concurrently, inference signals carry higher-order semantics at varying granularities, necessitating fine-grained integration with low-level multimodal features. These limitations reveal a fundamental gap: existing systems lack both reliable semantic reasoning and effective multimodal fusion mechanisms for hateful video detection.

To bridge this gap, we propose Reasoning-Aware Multimodal Fusion (RAMF), a unified framework that addresses both challenges. To tackle Challenge 1, we introduce contrastive reasoning—a structured three-stage process where a VLM generates (i) objective descriptions, (ii) hate-assumed inferences, and (iii) non-hate-assumed inferences—providing complementary semantic perspectives that enrich the model's contextual understanding while maintaining factual grounding. Unlike prior reasoning approaches (Lang et al., 2025; Hee et al., 2024; 2025), this contrastive design forces the model to explicitly consider both interpretations, providing complementary perspectives that enrich contextual understanding while maintaining factual grounding. To tackle Challenge 2, we design Local-Global Context Fusion (LGCF) to capture both local salient cues and global temporal structures, and propose Semantic Cross Attention (SCA) to enable fine-grained multimodal semantic interaction.

Our main contributions are: 1) Contrastive reasoning for nuanced and intent-aware semantic understanding. We propose a structured contrastive reasoning pipeline that generates objective descriptions, hate-assumed interpretations, and non-hate-assumed interpretations. This design provides complementary semantic views, enabling the understanding of nuanced hateful content in subtle contexts, improving robustness to nuanced scenarios. 2) LGCF and SCA for comprehensive multimodal fusion. We introduce LGCF to jointly capture local salient cues and global temporal patterns, and propose SCA to achieve fine-grained multimodal semantic interaction. This facilitates efficient fusion of heterogeneous modalities whilst integrating high-level reasoning signals. 3) Extensive experiments on HateMM and MultiHateClip demonstrate state-of-the-art performance, with improvements of 3% in Macro-F1 and 7% in hate class recall.

## 2 Related Work

### 2.1 Hateful Content Detection

For hate speech detection, early studies used manually designed text features. Chen et al. (2012) combined n-grams with grammatical rules, while Davidson et al. (2017) utilised sentiment analysis. Advances in deep learning have enabled automatic feature extraction, using convolutional neural networks (CNNs) (Le Cun et al., 1989) and recurrent neural networks (RNNs) (Elman, 1990) to identify hate patterns in text (Vashistha & Zubiaga, 2021; Menini et al., 2019; Corazza et al., 2020). Beyond text, hateful content also appears in multimodal forms (e.g., emojis and videos), necessitating the integration of multimodal methods. Das et al. (2023) developed the first hateful video dataset, HateMM, and proposed a standard paradigm for extracting multimodal information from video frames, audio, and transcribed text to detect hateful videos. Although recent models (Zhang et al., 2024; Lang et al., 2025; Céspedes-Sarrias et al., 2025; Rehman et al., 2025) incorporate multimodal information, they largely adhere to a standard fusion without reasoning or direct reasoning and fall short in capturing the nuanced, context-dependent nature of hateful content. For instance, while MoRE (Lang et al., 2025) leverages the BLIP visual language model to generate video frame descriptions, these captions remain surface-level and lack deeper semantic understanding of hate-related context.

On the other hand, the successful application of VLMs in multimodal tasks (Tang et al., 2025) has sparked interest in using them for hateful content moderation (Hee et al., 2024; Hee & Lee, 2025; Hee et al., 2025; Rizwan et al., 2025). For example, IntMeme (Hee & Lee, 2025) fine-tunes VLM models using carefully designed prompts and contrastive learning objectives. InstructMemeCL (Hee et al., 2025) uses VLM to generate human-style explanations for memes and then uses CoT to generate more transparent VLM decision results. These works demonstrate the potential of VLM for hateful content, but there has been no exploration of complex contextual semantic understanding in hateful videos. Given this research gap, we have redesigned a structured VLM generation system to generate semantic explanations, thereby aiding comprehension of contextually hateful content within nuanced contexts.

## 2.2 Multimodal Fusion in Hateful Video Detection

Das et al. (2023) developed the first hateful video dataset and established a baseline using a simple fusion method. Zhang et al. (2024) improved performance by employing complex cross-attention fusion techniques. Koushik et al. (2025) point out that existing fusion strategies have significant limitations in capturing complex multimodal semantic relationships and providing adaptable unified architectures. The latest work by Lang et al. (2025) enhanced hateful video detection performance through retrieval, expert fusion, and BLIP-based video description generation. However, the expert form of fixed modalities in it limits deep modal interaction and lacks multimodal semantic fusion.

Specifically for modelling and fusion methods, existing hateful video detection methods typically employ sequence models (e.g., long short-term memory networks (LSTM)) (Hochreiter & Schmidhuber, 1997) combined with attention mechanisms (Vaswani et al., 2017) to achieve modality interaction (Bai et al., 2018; Zhang et al., 2024; Das et al., 2023; Vashistha & Zubiaga, 2021; Mandal et al., 2024; Koushik et al., 2025). However, LSTM lacks effective local modelling capabilities, while traditional attention mechanisms (Vaswani et al., 2017), equipped with independent attention heads, lack direct interaction between heads. Recently, Multi-Token Attention (MTA) (Golovneva et al., 2025) addressed the head interaction issue by introducing key query convolutions and intra-group head mixing convolutions, thereby enabling information sharing among tokens. However, it mainly focused on contextual localisation and still lacked mechanisms to capture multimodal semantic structural dependencies. To address the above issues, we designed an improved attention module to achieve multi-semantic fusion of VLM-generated reasoning and modalities in traditional paradigms.

## 2.3 Semantic Modeling and Reasoning

Recent advances in video understanding have explored various strategies for semantic modeling under limited supervision. Existing works largely rely on implicitly learning semantic structures from visual features. For example, some approaches exploit local feature similarity to derive pseudo supervision, assuming temporally adjacent segments with high similarity share consistent semantics (Wang et al., 2026). Others enforce semantic consistency in representation space through contrastive learning, encouraging intra-class compactness and inter-class separability (Wang et al., 2024b). Beyond feature-level semantics, recent studies have investigated higher-level behavioural semantics such as intention, aiming to distinguish visually similar but semantically different events. These methods typically infer intention implicitly from temporal dynamics or motion patterns, highlighting the importance of abstract semantic understanding beyond appearance-level cues (Wang & Zhao, 2026). Other work in LLM safety has also explored leveraging opposing instructions for safety alignment through contrastive prompting or decoding (Zhong et al., 2024), where token probabilities induced by a positive prompt are adjusted by subtracting those from a negative prompt. Such approaches operate at the distribution level and produce a single output sequence without explicitly comparing generated reasoning. As a result, they lack the ability to model and verify alternative semantic interpretations at the reasoning level, limiting their effectiveness in capturing nuanced and context-dependent semantics in complex multimodal scenarios.

In parallel, the application of VLMs to hate speech detection has made some progress (Hee et al., 2025; Hee & Lee, 2025; Wang et al., 2025a; Rizwan et al., 2025; Lang et al., 2025; Yang et al., 2026). However, the field of hateful video detection has still not achieved significant breakthroughs. Currently, VLM-based methods typically generate a single narrative for a given piece of content, use VLMs to determine whether memes contain hate speech (Hee & Lee, 2025), or employ CoT to provide inferential explanations (Hee et al., 2025). However, recent research (Barez et al., 2025) points out that CoT is not explainable, indicating that CoT cannot guarantee the model's fidelity to the reasoning process, thereby lacking explainability. The infidelity of chain reasoning poses a trust crisis for hateful video detection systems. VLMs may make judgments based on implicit biases or incorrect context, yet generate seemingly reasonable but erroneous explanations to mask their true decision-making mechanisms (Barez et al., 2025), failing to meet legal requirements for algorithmic transparency and fundamentally threatening the credibility and controllability of hate content detection.

From a broader perspective, these approaches lie along a spectrum of semantic modeling: from implicit feature-based semantics, to intention-aware temporal modeling, and to language-based reasoning. Motivated by their limitations, we propose a structured contrastive reasoning framework that introduces explicit, fact-grounded semantic signals, enabling more reliable contextual understanding and multimodal interaction.

## 3 Methodology

### 3.1 Problem Formulation

Following the standard pipeline proposed in HateMM (Das et al., 2023), a video instance is represented by three modalities: video frames $V$, audio signal $A$, and transcription text $T$. Each modality $m \in \{T, A, V\}$ is encoded as an embedding sequence $X^m \in \mathbb{R}^{L \times D_m}$, where $L$ denotes the sequence length and $D_m$ denotes the feature dimension. The goal of hateful video detection is to learn a function $f(X^T, X^A, X^V) \to y$, where $y \in \{0, 1\}$ indicates whether the video contains hateful content.

However, as shown in Figure 1, the prior approach faces two fundamental challenges. The first is nuanced contextual understanding, where hateful intent is often expressed through nuanced multi-modal interactions and contextual associations. To address this issue, recent research has incorporated reasoning generated by VLMs into hate video detection, treating it as an auxiliary text modality that is either directly concatenated with the text modality or fused via a flattened scheme (Lang et al., 2025; Hee et al., 2024). Such methods treat reasoning as a homogeneous semantic signal, failing to distinguish its role from that of low-level modality features. This further exacerbates the second challenge of multimodal semantic fusion, namely the inability to effectively model fine-grained cross-modal semantic relationships. This calls for a more structured modeling of reasoning to explicitly capture diverse semantic roles and integrate them more effectively with multimodal features.

Therefore, we reformulate the problem as learning an enhanced function $f'$ that can leverage additional semantic signals to improve both contextual understanding and multimodal fusion, ultimately leading to more accurate prediction of hateful content.

### 3.2 Overview of the Proposed Framework

To address the above challenges, we propose a Reasoning-Aware Multimodal Fusion (RAMF) framework. Given an input video, the framework first extracts multimodal features from transcription text $T$, audio signals $A$, and video frames $V$, forming the basic semantic representation of the video. To enhance contextual understanding, we introduce structured reasoning signals generated by a VLM. Specifically, instead of relying on a single reasoning narrative, we design a contrastive reasoning scheme consisting of three components: an objective description ($T_O$), a hate-assumed interpretation ($T_H$), and a non-hate interpretation ($T_N$). The objective description provides a fact-grounded and neutral representation of the video content, supporting reliable reasoning. In contrast, the hate and non-hate interpretations form a pair of contrastive semantic hypotheses, enabling the model to reason about subtle and ambiguous contexts.

To effectively integrate these signals, we adopt a two-stage fusion strategy. First, the modalities with objective grounding $\{T, A, V, T_O\}$ are fused to obtain an intermediate representation $Y_1$, which establishes a grounded semantic basis. Then, the contrastive reasoning signals $\{T_H, T_N\}$ are incorporated to refine the representation into $Y_2$, enabling contrastive and discriminative reasoning for classification. The overall framework aims to learn a function $f'(X^T, X^A, X^V, X^{T_O}, X^{T_H}, X^{T_N}) \to y$, which integrates multimodal features with contrastive reasoning signals for hateful video detection. An overview of the framework is shown in Figure 2.

### 3.3 Vision Language Model Contrastive Reasoning

To address the limitations of existing VLM-based reasoning approaches, we propose a structured three-stage contrastive reasoning pipeline that enhances contextual semantic understanding without relying on VLM's subjective reasoning explanations. Unlike single-narrative approaches, our reasoning explicitly guides the VLM through a space of contrasting assumptions, generating complementary evidence for both hateful

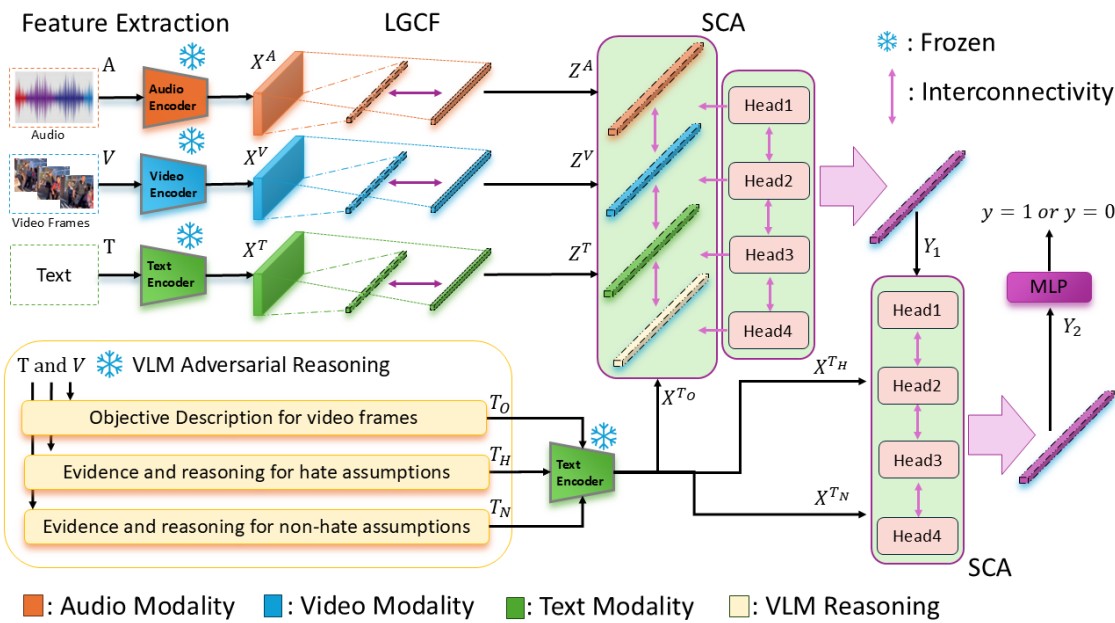

Figure 2: The overall architecture of the proposed framework, including the Local-Global Context Fusion (LGCF) module, the Semantic Cross Attention (SCA) mechanism.

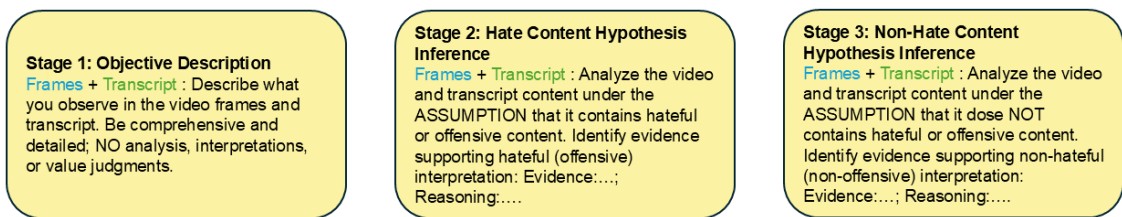

Figure 3: Prompts used in the three stages of our framework.

and non-hateful interpretations within the same video. This design enhances contextual understanding through three mechanisms: 1) the structured prompting constrains the VLM to produce objective descriptions before interpretive reasoning, reducing hallucination and bias; 2) the contrastive format provides self-correction—even if one reasoning path is flawed, the complementary perspective can compensate; and 3) explicit instructions requiring visual evidence references strengthen factual grounding, enhancing the reliability of model outputs. The robustness to variations in VLM quality and the impact of reasoning quality are validated in our ablation study.

The three stages include (1) Objective description of content: The model generates an objective description of the visual elements observed in the video and the accompanying text, without involving interpretative judgments, to establish an unbiased representation (see Figure 3), denoted as $T_O$.

(2) Hate-Assumed Inference: Assuming the content contains hate speech, the model explores discriminatory expressions and offensive content targeting specific groups, and provides contextual evidence and reasons (see Figure 3), denoted as $T_H$.

(3) Non-Hate-Assumed Inference: Assuming the content does not contain hate speech, the model explores reasonable alternative interpretations, such as artistic expression, satirical context, and personal conflicts, and provides corresponding contextual evidence and reasoning (see Figure 3), denoted as $T_N$.

### 3.4 Encoder Module

The encoder module comprises modality-specific preprocessing and feature extraction. Text, $T$, is obtained by transcribing speech with OpenAI's Whisper model (Radford et al., 2023), then tokenised and processed using Bidirectional Encoder Representations from Transformers (Bert or multilingual Bert (mBert)) (Devlin et al., 2019) and the HateXplain (HXP) model (Mathew et al., 2021) to extract 768-dimensional embeddings, with zero-padding or truncated to 100 fixed sequence length, denoted as $X^T$. The VLM reasoning text, $T_O, T_H, T_N$ also processed with Bert or HXP to get $X^{T_O}, X^{T_H}$ and $X^{T_N}$, then pass individually Multi-Layer Perceptron (MLP) with $\{512, 256\}$ to obtain unified dimensionality.

Audio, $A$, is resampled to 16kHz to extract 40-dimensional Mel Frequency Cepstral Coefficients (MFCC) (Muda et al., 2010), and to 48kHz to extract 512-dimensional semantic embeddings using the pretrained Contrastive Language-Audio Pretraining (CLAP) model (Elizalde et al., 2023), with zero-padding if needed or downsampling with 100 fixed stride, denoted as $X^A$.

Video frames, $V$, are processed by Vision Transformer (ViT) (Dosovitskiy et al., 2020) or Video Vision Transformer (Vivit) (Arnab et al., 2021) to extract 768-dimensional visual embeddings, and by the pretrained Contrastive Language-Image Pretraining (CLIP) model (Radford et al., 2021) to extract 512-dimensional semantic embeddings, with black frames padded if needed, denoted as $X^V$.

### 3.5 Local-Global Context Fusion

Based on observations of the hateful video, hateful content can be either concentrated in a short period of time or spread throughout long videos (Yang et al., 2025). This highlights the importance of local modelling and global understanding for traditional three modalities. However, existing LSTM-based methods have weak local modelling capabilities (Das et al., 2023; Zhang et al., 2024). Inspired by Bai et al. (2018), who studied the advantages of convolution for sequence modelling, the proposed LGCF module addresses issues by adaptively combining local salient features and global context within the sequence, while preserving discriminative cues and overall temporal structure, as illustrated in Figure 4.

Given the modality embeddings $X^m$ obtained from the encoders described in Section 3.3, where $m \in T, A, V$ denotes text, audio, and video modalities respectively (i.e., $X^T$, $X^A$, $X^V$ are the outputs of the text, audio and video encoders), we first apply modality-specific MLP with $\{512/128, 256\}$ to obtain the unified representation $X^m_{MLP}$ (128 with MFCC). Subsequently, $X^m_{MLP}$ is fed into two parallel channels to extract local and global temporal features. In the Local Temporal Channel (LTC), a one-dimensional convolution with kernel size 3 and padding 1 is applied across the time dimension to extract local context. The convolution kernel size is a non-critical parameter, as analysed in Figure 6. Then, maximum pooling is performed on the time axis to capture the maximum activation value of each feature:

$$v_{\text{local}} = \text{MaxPool1D}(\text{Conv1D}(X^m_{\text{MLP}})) \tag{1}$$

In the Global Temporal Channel (GTC), a global average pooling over the original sequence is computed:

$$v_{\text{global}} = \text{AdapAvgPool1D}(X^m_{\text{MLP}}) \tag{2}$$

The local and global representations are then combined through a gating mechanism to produce the fused modality representation $Z^m$:

$$g = \sigma \left( W \left( v_{\text{local}} \oplus v_{\text{global}} \right) + b \right),$$
$$Z^m = g \odot v_{\text{local}} + (1 - g) \odot v_{\text{global}} \tag{3}$$

where $\sigma$ denotes the sigmoid activation function, $v_{\text{local}}, v_{\text{global}} \in R^D$, $W \in R^{D \times 2D}$ and $b \in R^D$ are learnable parameters of the gating function, $\oplus$ denotes vector concatenation, $g \in R^D$ is a gating vector applied element-wise, and $\odot$ denotes element-wise multiplication. This design is crucial for detecting sparsity and implicit hate speech, ultimately compressing $X^m$ for each modality into a compact and information-rich representation $Z^m$, where $Z^m$ denotes the final fused representation for each modality $m$.

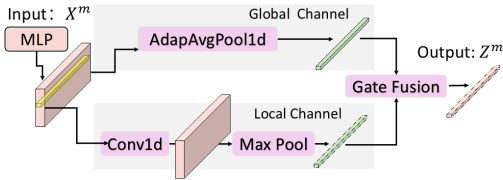

Figure 4: Structure of the LGCF, fusing local and global contextual information.

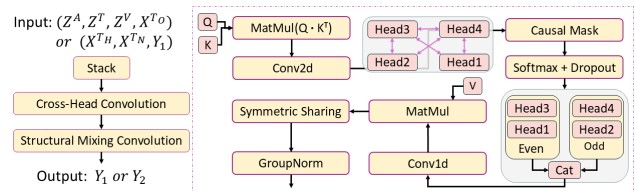

Figure 5: Structure of the SCA.

## 3.6 Semantic Cross Attention

Inspired by cross-head interactions in MTA (Golovneva et al., 2025), we propose the SCA mechanism. SCA introduces Cross-Head Convolution (CHC) and Structural Mixing Convolution (SMC) to facilitate comprehensive and fine-grained multimodal semantic fusion, as illustrated in Figure 5. To enable communication between heads and model the spatial structure of the query–key attention matrix, we apply 2D convolutions to the attention logits. Specifically, we treat the attention tensor as a 3D array of shape $[H, N, D^Z]$, where $H$ is the number of attention heads and $N$ is the sequence length. Each $[N \times D^Z]$ slice corresponds to an attention map for a single head. By treating the heads as convolution channels, we perform shared 2D convolutions. Unlike MTA (Golovneva et al., 2025), which assigns independent convolution kernels to each head, we apply a single shared convolution to all heads, achieving effective fusion of local information without additional parameter increases or modifications to the underlying operators, while reducing inference time, as analysed in Table 5.

The features from different modalities are stacked to form $Z^s \in R^{B \times 3/4 \times D}$, which is then fed into the SCA module ($Z^s$ will be omitted in the following text). CHC are employed to effectively capture high-order correlations across modalities,

$$A = \text{softmax}\left(\text{Conv2D}_{\text{heads}}\left(\frac{Q_h K_h^T}{\sqrt{d_h}}\right) + M\right) \tag{4}$$

where $Q_h$, $K_h$ and $d_h$ are the query, key matrices and dimensionality of each attention head, respectively. The causal mask $M$ is added to prevent access to future information. A $3 \times 3$ symmetric convolution operation with padding $1 \times 1$, denoted by $\text{Conv2D}_{\text{heads}}$. The $3 \times 3$ convolution kernel size used here is to accommodate the length of the input sequence $Z^s$, which is mostly 3.

$$\begin{aligned} A' &= \text{HeadMix}(A) \\ &= \text{Conv1D}_{\text{groups}=N/2}(A_{\text{even}} \oplus A_{\text{odd}}) \end{aligned} \tag{5}$$

For SMC, the attention weights $A$ are split into even- and odd-indexed heads, denoted as $A_{\text{even}}$ and $A_{\text{odd}}$. The odd-even grouping topology achieves interleaving and mixing between heads of different distances, improving the robustness and generalisation ability of the model, as analysed in Table 2. Then, concatenate to the sequence dimension, represented as $\oplus$. Group convolution $\text{Conv1D}_{\text{groups}=N/2}$ is used to process mixed views, similar to MTA (Golovneva et al., 2025). The default setting for the convolution kernel size is 2 with a stride of 2. The performance results for different sizes are basically stable, as analysed in Figure 6. Finally, the output of the module is as follows:

$$Y = \text{GN}\left(\text{OutProj}\left(A'V\right)\right) \tag{6}$$

where $V_h$ is the value matrix for each attention head. $A'V$ denotes the application of shared attention weights to the value matrix. The result is reshaped back to the original layout, producing the mixed attention weights $A'$. This is to extend the attention distribution after multimodal interaction to the entire attention space, considering structural alignment and semantic consistency. More analysis can be seen from the ablation study. Then followed by an output projection layer, OutProj, and group normalisation, GN.

For the SCA of the first layer, $Z^s \in \{X^T, X^A, X^V, X^{T_O}\}$, and get $Y_1$. For second layer SCA, $Z^s \in \{Y_1, X^{T_H}, X^{T_N}\}$, and get $Y_2$. For classification, $Y_2$ or $Y_1$ passes through an average pooling layer and

Table 1: Video classification performance (%) (five-fold average). Standard deviations are also reported in percentage points. MF1: Macro-F1; Acc: Accuracy; P(H): Precision for hate class; R(H): Recall for hate class. The MF row corresponds to a configuration without VLM inference.

| Model | HateMM | | | | MHC(Chinese) | | | | MHC(English) | | | |
|---|---|---|---|---|---|---|---|---|---|---|---|---|
| | MF1 | Acc | P(H) | R(H) | MF1 | Acc | P(H) | R(H) | MF1 | Acc | P(H) | R(H) |
| *Unimodal* | | | | | | | | | | | | |
| BERT[T1] | 78.6±2.0 | 79.5±1.4 | 74.3±2.3 | 74.9±6.9 | 58.8±4.8 | 63.1±4.0 | 44.8±6.2 | 49.2±14.5 | 62.6±1.1 | 65.1±1.4 | 49.4±2.3 | 58.6±7.1 |
| HXP[T2] | 80.6±1.3 | 81.2±1.4 | 74.7±3.2 | 80.8±3.8 | – | – | – | – | 64.0±1.4 | 67.8±1.4 | 54.0±3.9 | 56.8±5.0 |
| MFCC[A1] | 66.9±2.1 | 67.7±2.7 | 58.9±3.8 | 67.1±8.4 | 54.4±4.6 | 60.3±3.0 | 38.8±6.1 | 39.2±14.7 | 47.8±1.9 | 58.4±2.3 | 31.2±2.3 | 21.0±11.8 |
| CLAP[A2] | 72.2±1.5 | 74.2±1.6 | 71.1±3.6 | 59.7±2.4 | 59.1±3.4 | 66.2±2.0 | 48.2±4.9 | 38.0±7.7 | 57.9±2.3 | 64.8±2.7 | 47.4±3.7 | 36.6±2.8 |
| ViT[V1] | 71.2±2.6 | 72.8±2.6 | 67.1±3.5 | 62.6±5.2 | 61.9±3.9 | 65.5±4.1 | 48.2±5.7 | 54.3±9.2 | 56.6±5.2 | 61.2±3.5 | 43.8±6.8 | 41.9±11.6 |
| CLIP[V2] | 73.4±2.8 | 74.6±3.1 | 69.5±4.9 | 66.4±2.7 | 61.2±2.1 | 65.6±3.0 | 48.1±5.0 | 48.8±3.7 | 63.6±2.3 | 67.3±3.2 | 52.2±4.8 | 52.7±2.9 |
| *LLM* | | | | | | | | | | | | |
| GPT-4o | 78.0±1.1 | 78.2±1.0 | 66.8±2.4 | 89.8±2.8 | 54.0±2.1 | 70.4±3.7 | 71.8±4.1 | 16.4±2.6 | 63.7±3.8 | 72.4±3.2 | 70.1±7.5 | 34.2±4.9 |
| Qwen-Max | 66.8±2.4 | 67.0±2.4 | 55.2±2.9 | 92.3±2.9 | 61.6±1.6 | 68.4±2.3 | 53.5±7.8 | 40.3±3.2 | 70.2±4.8 | 73.0±4.3 | 60.6±7.7 | 62.3±6.8 |
| LLaMA 3.1 | 72.1±3.1 | 72.7±2.9 | 64.5±4.7 | 73.9±4.1 | – | – | – | – | 69.1±3.5 | 74.3±2.8 | 67.2±8.2 | 49.1±5.6 |
| *VLM* | | | | | | | | | | | | |
| Qwen-VL | 70.3±1.8 | 70.3±1.8 | 58.2±2.9 | 90.8±2.9 | 60.8±2.9 | 70.3±3.6 | 59.8±7.5 | 32.4±4.5 | 71.3±3.1 | 75.7±2.7 | 69.4±3.6 | 53.3±7.2 |
| LLaMA 4 | 69.8±3.3 | 69.8±3.2 | 57.8±4.1 | 91.2±1.6 | – | – | – | – | 69.4±2.7 | 74.4±2.3 | 66.9±3.9 | 50.2±5.5 |
| *Multimodal* T1·A1·V1 | | | | | | | | | | | | |
| HateMM | 79.3±2.0 | 79.8±2.2 | 73.9±2.4 | 79.5±5.6 | 64.0±2.7 | 68.3±2.3 | 51.8±3.3 | 53.1±11.6 | 62.3±5.1 | 67.6±5.0 | 54.0±10.6 | 44.0±5.8 |
| CMFusion | 79.1±1.7 | 80.2±2.0 | 77.1±4.0 | 72.2±5.0 | 61.4±2.2 | 68.6±2.1 | 53.5±6.1 | 39.3±5.2 | 60.8±2.2 | 66.8±3.1 | 52.3±4.9 | 40.6±2.2 |
| MoRE | 81.0±2.4 | 82.1±2.3 | 80.0±4.1 | 73.5±2.3 | 60.2±2.1 | 69.6±2.4 | 57.1±2.8 | 31.7±5.7 | 62.8±3.4 | 67.5±3.6 | 51.6±6.1 | 47.8±3.2 |
| MF | 82.3±3.1 | 82.9±3.2 | 78.0±3.8 | 80.2±3.9 | 65.9±2.0 | 70.3±2.8 | 55.1±2.6 | 53.2±5.8 | 63.5±4.3 | 67.8±4.3 | 53.3±5.8 | 50.9±12.0 |
| RAMF (Ours) | **83.7**±2.8 | **84.3**±2.7 | **78.6**±3.7 | **83.7**±4.5 | **69.3**±2.3 | **72.4**±4.2 | **58.4**±6.0 | **63.8**±9.2 | **64.1**±6.2 | **68.5**±3.1 | **53.2**±5.6 | **52.2**±16.9 |
| T2·A2·V2 | | | | | | | | | | | | |
| HateMM | 82.4±2.4 | 83.0±2.6 | 79.4±4.1 | 80.2±5.1 | 63.3±6.1 | 69.0±4.5 | 54.3±7.5 | 47.3±15.6 | 68.4±4.7 | 72.3±2.1 | 59.1±5.9 | 57.1±12.9 |
| CMFusion | 81.9±2.1 | 82.8±2.0 | 80.2±2.6 | 75.8±4.1 | 60.6±3.8 | 66.7±3.2 | 49.0±4.7 | 42.4±7.7 | 67.7±2.7 | 71.7±2.2 | 60.6±6.6 | 55.1±1.0 |
| MoRE | 82.1±2.1 | 82.9±2.4 | 77.0±3.4 | 79.7±5.6 | 62.5±4.5 | 69.1±4.3 | 54.1±7.1 | 41.2±12.1 | 67.4±3.5 | 72.5±3.1 | 61.1±8.5 | 49.3±13.4 |
| MF | 83.2±3.3 | 83.7±3.5 | 78.3±6.0 | 82.6±2.9 | k6.2±4.4 | 70.3±4.1 | 55.2±8.1 | 54.9±8.4 | 69.6±4.3 | 73.4±3.3 | 62.1±7.8 | 56.8±9.0 |
| RAMF (Ours) | **85.1**±1.9 | **85.6**±1.8 | **79.8**±1.9 | **85.5**±5.4 | **70.9**±4.3 | **74.5**±4.0 | **61.3**±6.4 | **62.2**±11.5 | **71.7**±4.3 | **74.0**±4.1 | **62.1**±9.3 | **67.4**±11.0 |

enters the MLP classification layer with $\{128, 64, 2\}$ to obtain the final classification result. For RAMF, use $Y_2$. For a traditional paradigm-based Multimodal Fusion (MF) model, use $Y_1$.

# 4  Experiments

A total of 1,083 HateMM videos and 959/964 videos from the Chinese/English MHC subsets are used. These numbers differ from the 1,000 videos per subset reported by Wang et al. (2024a) as we re-collected and re-partitioned the data to construct a more rigorous five-fold cross-validation setup with mutually exclusive test sets. During this process, some original videos were found to be unavailable due to removals from Bilibili and YouTube, resulting in slight deviations in dataset size. We adopt a 70%/10%/20% split for train/validation/test within each fold and perform 5-fold cross-validation. Unlike prior work that fixed a single data split and only varied random seeds across runs (Lang et al., 2025; Wang et al., 2024a), our re-partitioning ensures that test sets across folds are strictly non-overlapping, enabling a more generalisable and realistic evaluation. In the binary classification task for MHC, hate and offensive labels are merged into a single hate label, following consistent practice in MHC. Our models are trained using the Adam optimiser with a learning rate of $10^{-4}$ and cross-entropy loss; more configuration details are provided in appendix D.

Following prior unified protocol (Das et al., 2023; Wang et al., 2024a), we sample an average of 100 frames per video in HateMM and 32 frames in MHC. For VLM reasoning, we employ the Qwen 2.5-VL-32B (Bai et al., 2025b), using 16 sampled frames per video as visual input. This configuration is selected to accommodate hardware constraints. For baseline models, we strictly adhere to the settings specified in their original papers. We maintain the same evaluation metrics used in the HateMM and MHC, including macro-F1 score, accuracy, F1 score for hate class, precision for hate class, and recall for hate class. The best model for each fold is selected based on the macro F1 score on the validation set and evaluated on the test set.

Table 2: Ablation study on the HateMM dataset using BERT, MFCC, and ViT. MF1 denotes macro-F1, and Acc denotes accuracy. Values in parentheses indicate the relative decrease in macro-F1 compared to the RAMF[1] or MF baseline. All values are reported in percentage (%), with standard deviations also expressed in percentage.

| RAMF Ablation | | | MF Ablation | | |
|---|---|---|---|---|---|
| | MF1 | Acc | | | |
| RAMF[1] | 84.26 | 83.75 | | | |
| RAMF[2] | 84.35 | 83.62 | | MF1 | Acc |
| RAMF[3] | 83.61 | 82.87 (↓0.88) | MF | 82.96 | 82.32 |
| MF-CoT | 83.24 | 82.61 (↓1.14) | w/o MLP | 80.83 | 80.08 (↓2.24) |
| w/o Hier. Fusion | 82.78 | 81.95 (↓1.80) | w/o LGCF | 80.74 | 79.87 (↓2.45) |
| w/o ObjDesc | 83.80 | 83.14 (↓0.61) | w/o SCA | 80.28 | 79.41 (↓2.91) |
| w/o Assumption | 82.41 | 81.69 (↓2.06) | | | |

| SCA Module Ablation | | | LGCF Module Ablation | | |
|---|---|---|---|---|---|
| | MF1 | Acc | | | |
| w/o CHC | 80.83 | 80.12 (↓2.20) | | MF1 | Acc |
| w/o SMC | 82.13 | 81.23 (↓1.09) | w/o Gate Fusion | 80.56 | 79.44 (↓2.88) |
| Concat | 81.94 | 81.11 (↓1.21) | w/o GTC | 79.63 | 78.87 (↓3.45) |
| MTA | 81.94 | 81.18 (↓1.14) | w/o LTC | 77.87 | 77.21 (↓5.11) |
| StdAttn | 79.07 | 77.89 (↓4.43) | LSTM | 77.96 | 76.82 (↓5.50) |
| CrossAttn | 78.89 | 78.28 (↓1.84) | | | |

### 4.1 Baselines

We evaluate the effectiveness of the proposed model by comparing it with recent unimodal and multimodal approaches on the HateMM and MHC datasets, which are two real-world video datasets in the field. For unimodal baselines, we adopt CLIP (Radford et al., 2021), ViT (Dosovitskiy et al., 2020) (Vivit (Arnab et al., 2021) in MHC), CLAP (Elizalde et al., 2023), MFCC (Muda et al., 2010), HXP (Mathew et al., 2021) and BERT (Devlin et al., 2019) (mBert in MHC–Chinese). We apply average pooling with an MLP to each unimodal input, following HateMM (Das et al., 2023) and MHC (Wang et al., 2024a). We additionally include LLMs and VLMs, i.e., GPT4–o (OpenAI, 2024), LLama (3.1–405B and 4–17B) (Patterson et al., 2022), and Qwen–Max (Qwen, 2024) for comparison by evaluating their zero-shot performance on hateful video classification. LLMs and VLMs are guided via prompts to analyse textual and video-text inputs; further details are given in the appendix B.5. For multimodal models, HateMM (Das et al., 2023) and MHC (Wang et al., 2024a) are used as baseline models, along with CMFusion (Zhang et al., 2024) and the state-of-the-art MoRE (Lang et al., 2025) model.

### 4.2 Quantitative Results

Table 1 shows the performance comparison between our proposed model and existing methods on hateful video detection. Our model achieves the best performance on all datasets and different feature combinations, demonstrating excellent robustness and generalisation ability. The MF row in Table 1 corresponds to a configuration without VLM inference, designed to demonstrate the advantages of the fusion module while still achieving significant improvements over previous fusion methods, validating the effectiveness of our proposed SCA and LGCF in multimodal semantic fusion. RAMF further improves performance, demonstrating that our contrastive reasoning and multimodal fusion design effectively enhances robustness to nuanced and context-dependent hate content. In particular, it consistently improves both macro-F1 and hate-class recall, which are critical for hateful video detection. RAMF further improves performance, demonstrating that our novel contrastive reasoning framework effectively enhances the model's robustness against nuanced and context-dependent hate content, particularly by simultaneously improving macro-F1 and recall, which are crucial for hate video detection. More ablation analysis demonstrating superiority over CoT methods.

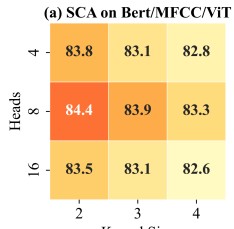 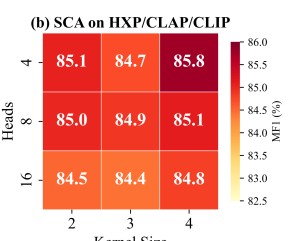 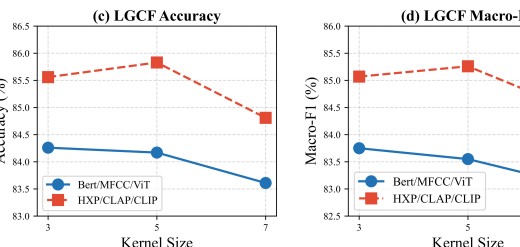

Figure 6: Hyperparameter analysis across two feature configurations on the HateMM dataset. Acc denotes accuracy, and MF1 denotes macro-F1.

Table 3: Performance comparison of different vision-language models (VLMs) used in the RAMF framework on the HateMM dataset with Bert, MFCC and Vit. All values are reported in percentage (%), with standard deviations also expressed in percentage.

| VLM | Acc | MF1 |
|---|---|---|
| Gemini-2.5-Flash | $84.17 \pm 1.80$ | $83.38 \pm 1.54$ |
| GPT-5-mini | $84.72 \pm 3.46$ | $83.96 \pm 3.14$ |
| Qwen2.5-VL-32B | $83.75 \pm 2.81$ | $84.26 \pm 2.72$ |
| LLaMA-4-17B | $83.62 \pm 2.91$ | $84.35 \pm 2.86$ |
| Qwen2.5-VL-7B | $82.69 \pm 3.92$ | $81.81 \pm 3.86$ |
| Qwen3-VL-2B | $82.59 \pm 2.87$ | $81.49 \pm 3.07$ |

## 4.3 Ablation Study

For the ablation study presented in Table 2, we analyse the individual contributions of each proposed component. In the RAMF ablation, we compare different strategies: RAMF[1] represents the proposed framework using Qwen 2.5-VL-32B, while RAMF[2] substitutes LLaMA 4-17B as the reasoning generator. The marginal performance difference between RAMF[1] and RAMF[2] demonstrates that our framework is robust to variations in VLM quality. RAMF[3] fuses the objective description in the second SCA layer instead of the first. The performance drop observed when using the CoT reasoning method (MF-CoT) instead of contrastive reasoning highlights the effectiveness of the latter (detailed CoT implementation is provided in the appendix B.3). Further, removing the second-layer SCA (w/o hierarchical fusion) and instead processing all information in a single SCA results in performance degradation. Similarly, excluding either the objective descriptions (w/o ObjDesc) or the contrastive reasoning (w/o Assumption) leads to reduced performance. Notably, eliminating the contrastive reasoning capability leads to a decline in MF1 by over 2%, demonstrating its significant role.

In the MF ablation study, removing any single module leads to a performance drop, validating the overall design and necessity of each component. More granular ablation of the SCA reveals that eliminating core mechanisms such as CHC or SMC reduces performance, proving the effectiveness of these mechanisms for semantic communication and integration. Comparisons with standard attention (StdAttn) (Vaswani et al., 2017) fusion mechanisms, cross attention (CrossAtten) (Tsai et al., 2019) fusion mechanisms, MTA (Golovneva et al., 2025) and replacing structured odd-even grouping with simple concatenation (Concat), further affirm the superiority of the enhanced attention architecture. Ablation results from the LGCF module confirm the necessity of both the gating mechanism and the dual channel structure. Additionally, replacing the LGCF with an LSTM architecture results in a performance decline, indicating that the proposed framework effectively captures local and global spatio-temporal features without relying on conventional sequential processing constraints.

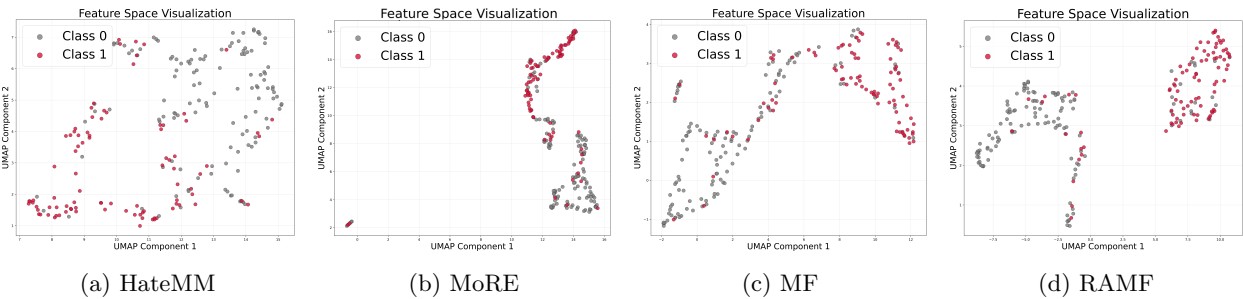

| (a) HateMM | (b) MoRE | (c) MF | (d) RAMF |

Figure 7: UMAP (McInnes et al., 2020) space visualisations across different models (HXP/CLAP/CLIP). Class 0: Non-hate. Class 1: Hate.

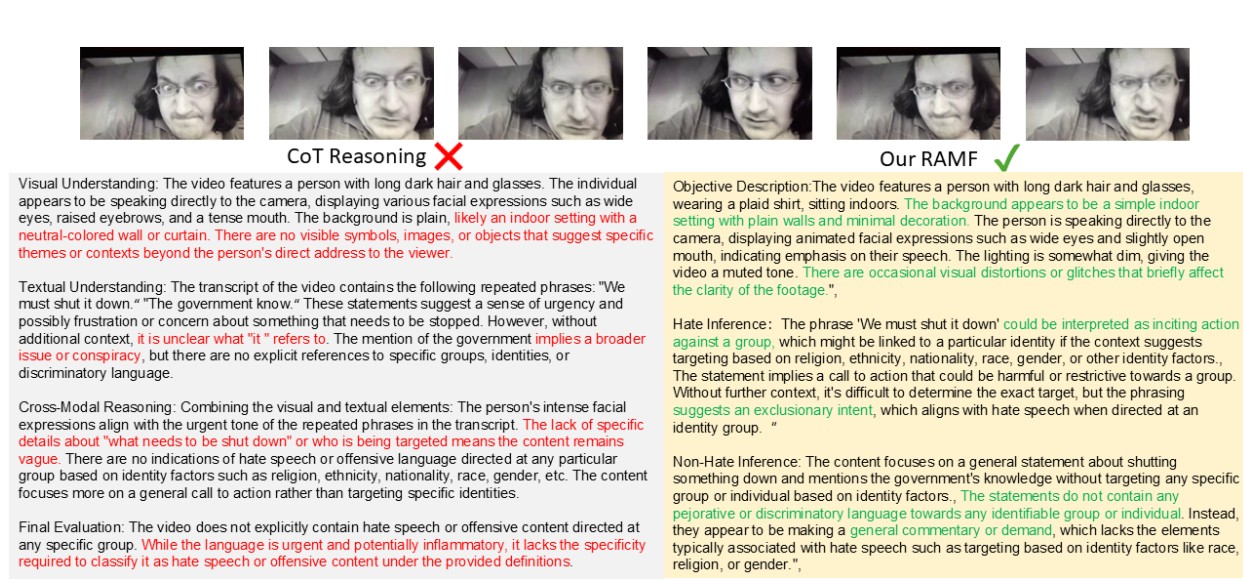

Figure 8: Comparison of reasoning strategies

## 4.4 Impact of Different Vision-Language Models

To investigate the influence of different VLMs, we replace Qwen2.5-VL-32B with several alternative models, including LLaMA-4-17B (AI, 2025), GPT-5-mini (OpenAI, 2025), Gemini-2.5-Flash(DeepMind, 2025), Qwen2.5-VL-7B (Bai et al., 2025b) and Qwen3-VL-2B (Bai et al., 2025a) . As shown in Table 3, the performance remains relatively stable across different VLMs, indicating that our framework is not sensitive to the choice of a specific high-capacity model. Moreover, advanced models such as GPT-5-mini and Gemini-2.5-Flash achieve competitive performance, suggesting that strong results can be obtained relying on a larger VLM. In contrast, the smaller 7B and 2B VLMs exhibit a noticeable performance drop, implying that insufficient reasoning capacity may limit the quality of generated semantic signals.

## 4.5 Hyperparameter Analysis

We analyse the sensitivity of RAMF to key hyperparameters, including the number of attention heads and convolutional kernel sizes in both SCA and LGCF modules (Figure 6). The results demonstrate that RAMF is relatively insensitive to moderate changes in these settings. The performance across different configurations remains stable, indicating the robustness of the proposed architecture.

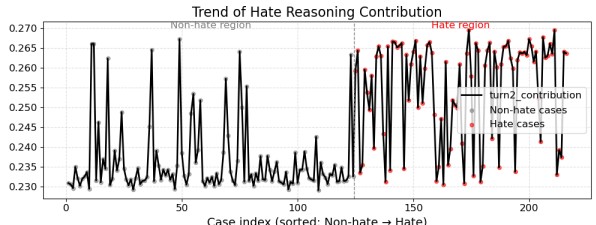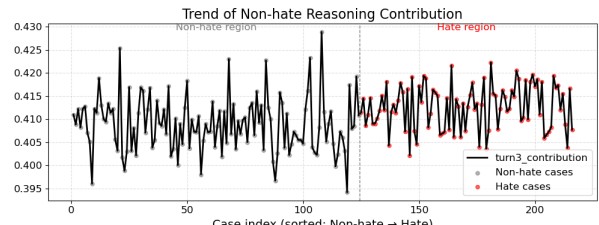

Figure 9: Trends of hate and non-hate reasoning contributions across the HateMM dataset. The left panel shows the variation in hate reasoning contribution, while the right panel illustrates the non-hate reasoning contribution. The y-axis represents the normalized contribution score of each information source, computed from the averaged attention matrix in the second SCA layer. The contributions are normalized to sum to 1. The dataset is sorted from non-hate to hate instances, with the right-side region corresponding to hate cases.

### 4.6 Qualitative Analysis

Figure 7 visualises the feature space. Compared with prior methods, the boundaries between hateful and non-hateful samples in the baseline feature space are blurred, whereas the distribution of RAMF embeddings is more compact and better separated.

To demonstrate the advantage of our contrastive reasoning design, Figure 8 presents a qualitative comparison between CoT reasoning and RAMF. As shown on the left, CoT produces long but weakly grounded explanations that rely heavily on speculative observations and the absence of explicit hateful cues. It fails to resolve ambiguous textual references (e.g., "it"), and its cross-modal analysis remains superficial, offering no mechanism to complete missing contextual information. In contrast, RAMF generates structured, contrastive reasoning that systematically explores both hateful and non-hateful interpretations. The objective description provides a neutral, hallucination-free account of the visual scene, while the hate-assumed and non-hate–assumed inferences offer complementary semantic hypotheses. This contrastive setup forces the model to surface potential identity-targeting implications (when present) and, equally importantly, to articulate legitimate non-hateful explanations when the evidence supports neutrality. As reflected in the figure, RAMF not only resolves ambiguous textual cues but also supplies balanced, evidence-grounded reasoning, enabling more reliable intent interpretation and reducing false positives driven by surface-level correlations.

To further analyse RAMF's behaviour across different types of instances, we plot the trend of reasoning contributions in Figure 9. Let $A \in R^{N \times N}$ denote the averaged attention matrix across heads in the second SCA layer. The contribution score of token $i$ is computed as

$$c_i = \text{softmax}\left(\sum_{j=1}^{N} A_{j,i}\right), \tag{7}$$

which measures how strongly token $i$ is attended to by other tokens. The left panel presents the hate reasoning contribution, showing a clear increase when entering the hate-case region (right side of the plot). In contrast, the right panel displays the non-hate reasoning contribution, which remains relatively stable across non-hate cases and slightly decreases in the hate region. This trend demonstrates that RAMF adaptively shifts the focus of its contrastive reasoning depending on whether the input contains hateful intent, aligning well with the expected behaviour of a robust hate-speech detection model.

Figure 10 demonstrates RAMF's ability to interpret nuanced multimodal cues and accurately infer intent across diverse scenarios. The HateMM model serves as the baseline because it represents the standard multimodal pipeline widely used in prior work. Whether hateful signals appear subtly in text, are embedded within neutral or analytical discussions, or are mixed with noisy or ambiguous visuals, RAMF consistently aligns linguistic, visual, and contextual information to recover the correct meaning. These examples highlight RAMF's strength in capturing fine-grained semantics and context-dependent cues, enabling robust and intent-aware understanding of complex video content.

| | Case 1 | Case 2 | Case 3 |
|---|---|---|---|
| |  |  |  |
| **Visual** | Static frames showing a vintage Columbia record label; no explicit hateful visual cues. | A speaker on a dim stage giving a talk; no aggressive or harmful actions depicted. | Black-and-white surveillance-style video showing a confrontation outside a house; tense scene with multiple individuals interacting. |
| **Text** | The song title contains a highly offensive racial slur, representing a *subtle but strong hate cue* that requires precise text understanding. | Mentions historically sensitive groups (e.g., Jewish communities), but used in *informational or analytical context*, not in a derogatory way. | Overlay text includes hateful and racially derogatory comments added in post-production, providing strong hate cues. |
| **Audio** | No meaningful speech content; audio is non-informative for classification. | Academic lecture tone; calm delivery with no hateful intent conveyed through prosody. | Urgent shouting and commands suggesting distress, aligning with the heated scenario. |
| **Ground True** | Hate | Non–Hate | Hate |
| **Baseline** | ✗ | ✗ | ✗ |
| **RAMF** | ✅ | ✅ | ✅ |

Figure 10: Representative cases comparing the baseline HateMM model and our proposed RAMF.

## 5 Conclusion

In this work, to tackle the challenges of nuanced context understanding and multimodal semantic fusion in hateful video detection, we propose a novel Reasoning-Aware Multimodal Fusion framework. This framework consists of two core components: (1) Contrastive Reasoning, which generates complementary hate/non-hate perspectives through a structured three-stage VLM process, providing contextually grounded semantic information that avoids the limitations of direct reasoning and CoT approaches. (2) A novel fusion mechanism comprising Local-Global Context Fusion that captures both local salient cues and global temporal structures, and Semantic Cross-Attention that enables fine-grained multimodal semantic interaction. Extensive experiments on two benchmarks demonstrate the strong detection ability and generalizability of RAMF, providing a promising solution to context-aware hateful video detection.

**Broader Impact Statement**

This work aims to improve the detection of hateful content in multimodal videos, contributing to safer online platforms and more effective content moderation. By enhancing context understanding and multimodal reasoning, the proposed framework may help reduce the spread of harmful speech and protect vulnerable communities. However, automated detection systems may still produce false positives or inherit biases from underlying models, potentially affecting fair moderation. We emphasise that such systems should be deployed with human oversight, bias evaluation, and transparent governance to ensure responsible use.

**Acknowledgments**

This work was supported by the Alan Turing Institute and DSO National Laboratories Framework Grant Funding.

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

# A  Dataset Details

We conduct experiments on two widely used hateful video datasets, HateMM and MHC, which differ substantially in language, annotation granularity, and multimodal characteristics. HateMM consists of 1,083 English videos collected from BitChute, spanning approximately 43 hours of content, while MHC is a multilingual benchmark containing approximately 2,000 short videos, evenly divided into English (YouTube) and Chinese (Bilibili) subsets (Das et al., 2023; Wang et al., 2024a).

Regarding annotation protocol, HateMM adopts a relatively standard binary labeling scheme in which each video is annotated independently by two annotators, and disagreements are resolved by an expert annotator. The reported inter-annotator agreement is Cohen's k = 0.625, indicating substantial agreement (Das et al., 2023). In contrast, MHC employs a more fine-grained and structured annotation pipeline. Each video is first annotated by two annotators, with disagreements resolved through an additional annotator and finalized via majority voting. The inter-annotator agreement is k = 0.62 for English and k = 0.51 for Chinese in the multi-class setting, which improves to k = 0.72 and k = 0.66, respectively, under binary classification (Wang et al., 2024a).

In terms of label distribution, HateMM follows a binary classification setting with approximately 39.8% hateful and 60.2% non-hateful videos, resulting in a relatively balanced dataset compared to many hate speech corpora (Das et al., 2023). MHC, on the other hand, adopts a three-class labeling scheme (hateful, offensive, and normal), with roughly 30% of videos labeled as hateful or offensive and 70% as normal (Wang et al., 2024a). Following prior work (Zhang et al., 2024; Wang et al., 2024a; Yue et al., 2025), we merge the hateful and offensive categories into a unified "hate" class for binary classification. Notably, MHC exhibits a clear imbalance in hate class.

The two datasets also differ significantly in temporal characteristics. HateMM contains relatively long-form videos with an average duration of approximately 2.4 minutes, where hateful content may appear sparsely or evolve over time. In contrast, MHC consists of short clips with average durations of approximately 34 seconds for English videos and 32 seconds for Chinese videos, where hateful cues are typically more temporally concentrated.

From a multimodal perspective, both datasets provide visual, audio, and textual information, but their modality completeness and reliability vary. In HateMM, all videos include transcripts generated via automatic speech recognition; however, the transcript quality is relatively noisy, with approximately 22% out-of-vocabulary words (Das et al., 2023). In contrast, MHC explicitly annotates modality contributions and reveals that hateful content is often inherently multimodal. For example, 80.4% of Chinese hateful videos involve multiple modalities jointly conveying hate signals, highlighting the importance of cross-modal interaction (Wang et al., 2024a).

# B  VLM Inference Experiment

## B.1  Large Language Models (Text-only)

Each English or Chinese transcript is sent to a Large Language Model (LLM) with the fixed prompt below:
*"Please determine whether the following English/Chinese text contains hateful content. If it contains hateful content, please return 1; if it does not contain hateful content, please return 0. Just return 1 or 0, no other words. Here is the following text:* {text}"

## B.2  Vision–Language Models (Image + Text)

For multimodal inference, we uniformly extracted five frames from each video. Although the Qwen-VL-Max (Qwen, 2024) model supports more than five images, the Llama 4 (Patterson et al., 2022) recommends a maximum of five images. To maintain consistency, we have standardised the input to five images. All five frames, in chronological order, are provided together with the transcript through the following prompt:
*"Please analyse both the video frames and the following text to determine if they contain hateful content. If*

*contain hateful content, please return 1; if not contain hateful content, please return 0. Just return 1 or 0, no other words. Here is the text:* {text} ″

The placeholder *{text}* is replaced by the raw transcript. The model therefore performs binary classification in a strict zero-shot setting and must output exactly "1" or "0".

### B.3 Implementation Details of Chain-of-Thought

Chain-of-Thought (CoT) prompting has become a widely used approach to improve reasoning quality in complex tasks. In this work, we design a structured CoT prompt to guide the model through a stepwise analysis of potentially harmful video content, as shown in Figure 11. The prompt explicitly breaks down the task into four stages—visual understanding, textual analysis, multimodal reasoning, and final evaluation—encouraging the model to consider different aspects of the content systematically before making a judgment.

This is a video that may contain harmful content such as hate speech, explicit violence, discrimination, or other offensive behavior. You are a content moderation expert. Analyze this video using a reasoning process. The video is represented by: Visual frames and Transcript text (could be none).

HATE SPEECH DEFINITION: Any communication that attacks or uses pejorative/discriminatory language targeting a person or group based on their religion, ethnicity, nationality, race, colour, descent, gender or other identity factor. OFFENSIVE CONTENT: Contains hurtful, derogatory, or socially inappropriate content, even if it does not constitute direct hate speech. IMPORTANT: Simple conflicts, arguments, insults, or violent behaviour between individuals may not constitute hate speech.

Please reason in four steps, then summarize your final judgment.
Step 1: Visual Understanding. Describe the visual content in the frames. Focus on characters, scenes, and potential symbolic or hateful imagery.
Step 2: Textual Understanding. Analyze the transcript of the video. Pay attention to metaphors, puns, homophones, or rhetorical devices that may express hate or implicit bias.
Step 3: Multimodal Reasoning. Integrate insights from the visuals and the transcript. Identify any \text{implicit meanings}, contradictions, or cross-modal cues that enhance or modify the hateful nature of the content.
Step 4: Final Evaluation. Does this video express harmful or hateful content? Explain in 1-2 sentences why. Keep the answer concise, structured (following these four steps), and professional.

Figure 11: Chain-of-Thought prompt.

### B.4 Implementation Details of Contrastive Reasoning

Figure 12 presents the full prompt template used to implement the contrastive reasoning procedure. The prompt is explicitly structured into three sequential stages: (1) objective visual description, which restricts the model to observable entities, actions, and textual content without interpretation; (2) hate-assumed inference, where the model is instructed to identify potential hateful or offensive signals under the assumption that such content is present; and (3) non-hate-assumed inference, which enforces an alternative interpretation assuming the absence of hate speech. Each stage defines clear task boundaries and output constraints, enabling consistent and reproducible generation of reasoning text that can be directly encoded and fused with multimodal features.

### B.5 Model Versions

Table 4: Model used in zero-shot evaluation.

| Category | Model identifier |
|---|---|
| LLM | GPT-4o-2024-05-13 |
| | Qwen-Max-2025-01-25 |
| | Llama-3.1-405B-Instruct |
| VLM | Qwen-VL-Max-2025-01-25 |
| | Llama-4-Maverick-17B-128E-Instruct |

Step 1: Objective Description
You are a content analysis expert. Provide a detailed, objective description of this video content.
Task: Describe what you observe in the video frames and transcript:
1. objective_visual_description: Describe what you see - objects, people, actions, settings, text, symbols, colors, expressions, gestures, etc.
Rules:
Be comprehensive and detailed;
NO analysis, interpretations, or value judgments;
Simply report what you observe;

Step 2: Hate-assumed Inferences
You are a hateful content detection specialist. Analyse the video and transcript content under the ASSUMPTION that it contains hateful or offensive content.
HATE SPEECH DEFINITION: Any communication that attacks or uses pejorative/discriminatory language targeting a person or group based on their religion, ethnicity, nationality, race, colour, descent, gender or other identity factor.
OFFENSIVE CONTENT: Contains hurtful, derogatory, or socially inappropriate content, even if it does not constitute direct hate speech.
IMPORTANT: Simple conflicts, arguments, insults, or violent behaviour between individuals may not constitute hate speech.
Task: Identify evidence supporting hateful(offensive) interpretation:
1. evidence: Describe specific evidence supporting the hateful nature of the content 2. reasoning: Explain why it is hateful content.
Focus on:
- Language targeting religion, ethnicity, race, gender, nationality, etc.
- Group-based dehumanizing language or symbols
- Identity-based discriminatory attacks (not personal insults)
- Incitement against identity groups

Step 3: Non-hate-assumed Inferences
You are a content analyst. Analyze this content under the ASSUMPTION that it does NOT constitute hate speech and offensive content.
HATE SPEECH DEFINITION: Any communication that attacks or uses pejorative/discriminatory language targeting a person or group based on their religion, ethnicity, nationality, race, colour, descent, gender or other identity factor.
OFFENSIVE CONTENT: Contains hurtful, derogatory, or socially inappropriate content, even if it does not constitute direct hate speech.
IMPORTANT: Simple conflicts, arguments, insults, or violent behaviour between individuals may not constitute hate speech.
Task: Identify evidence supporting non-hate(non-offensive) interpretation:
1. evidence: Describe specific evidence supporting the non-hateful nature of the content
2. reasoning: Explain why this content does not hateful
Consider:
- Is this a personal dispute rather than group targeting?
- Are insults directed at individuals rather than identity groups?
- Is there artistic, satirical, or educational context?
- Does the content lack group-based discriminatory language?

Figure 12: Contrastive reasoning prompt.

Table 4 lists the models evaluated in this study, covering both large language models (LLMs) and vision-language models (VLMs). The LLMs include GPT-4o (OpenAI, 2024), Qwen-Max(Qwen, 2024), and Llama-3.1 (Patterson et al., 2022), while the VLMs include Qwen-VL-max (Qwen, 2024) and Llama-4-Maverick (Patterson et al., 2022). GPT-4o inferences were conducted via the official OpenAI API, and all Qwen and Llama variants were accessed through Alibaba Cloud's generative-AI service.

## C  End-to-End Implementation of the RAMF Framework

Algorithm 1 summarises the end-to-end training and inference procedure of the proposed RAMF framework. For each input video instance, the pipeline first invokes a vision–language model to generate three types of reasoning texts—objective description, hate-assumed inference, and non-hate-assumed inference—following the predefined contrastive prompting strategy. Multimodal features are then extracted independently from text, audio, video, and reasoning outputs using modality-specific encoders and unified through lightweight MLP projections. The fused representations are processed by the Local–Global Context Fusion module to capture complementary temporal cues, followed by a hierarchical Semantic Cross Attention mechanism that integrates both low-level modalities and high-level reasoning signals. Finally, the aggregated representation is passed to a classifier to produce the hate/non-hate prediction.

---

**Algorithm 1** Training of RAMF for hateful video detection.

---

**Input:** The hateful video dataset $\mathcal{S} = \{S_1, \cdots, S_N\}$.
**Output:** Predicted category $\hat{y}$ (Hate or Non-hate).

1: **for each** instance $S_i$ in $\mathcal{S}$ **do**
2:     */* VLM Contrastive Reasoning */*
3:     Generate objective description $T_O$ using VLM with video frames and transcript of $S_i$.
4:     Generate hate-assumed inference $T_H$ under the assumption that the content contains hate speech.
5:     Generate non-hate-assumed inference $T_N$ under the assumption that content does not contain hate speech.
6:     */* Feature Extraction */*
7:     Extract features $X^m$, $m \in \{a, v, t\}$ from audio, video, and text modalities of $S_i$ using MFCC/CLAP, ViT/CLIP, and BERT/HXP, respectively.
8:     Encode reasoning texts $T_O$, $T_H$, $T_N$ using text encoder to obtain $X^{T_O}$, $X^{T_H}$, $X^{T_N}$.
9:     */* Local-Global Context Fusion (LGCF) */*
10:     **for** $m \in \{t, a, v, T_O\}$ **do**
11:         Apply MLP to $X^m$ to obtain unified representation $X^m_{\text{MLP}}$.
12:         Extract local features: $v_{\text{local}} = \text{MaxPool1D}(\text{Conv1D}(X^m_{\text{MLP}}))$.
13:         Extract global features: $v_{\text{global}} = \text{AdapAvgPool1D}(X^m_{\text{MLP}})$.
14:         Compute gating weight $g = \sigma(W[v_{\text{local}} \oplus v_{\text{global}}] + b)$.
15:         Fuse features: $Z^m = g \odot v_{\text{local}} + (1 - g) \odot v_{\text{global}}$.
16:     **end for**
17:     */* First-Layer Semantic Cross Attention (SCA) */*
18:     Stack the modality features: $Z_s^{(1)} = [Z^t, Z^a, Z^v, Z^{T_O}]$.
19:     Apply SCA with Cross-Head Convolution (CHC) and Structural Mixing Convolution (SMC) to obtain $Y_1$.
20:     */* Second-Layer Semantic Cross Attention */*
21:     Encode contrastive reasoning into feature space and stack: $Z_s^{(2)} = [Y_1, X^{T_H}, X^{T_N}]$.
22:     Apply SCA to $Z_s^{(2)}$ to obtain final representation $Y_2$.
23:     */* Classification */*
24:     Apply average pooling to $Y_2$: $Y_{\text{pool}} = \text{AvgPool}(Y_2)$.
25:     Feed $Y_{\text{pool}}$ into MLP classifier to obtain prediction $\hat{y}_i$.
26: **end for**

---

## D  Additional Experimental Details

We re-partition the five-fold dataset such that the test sets across all five folds are mutually exclusive—each test set contains a distinct subset of the data. While training and validation splits may partially overlap across folds, this design ensures that the model is evaluated on entirely different test data in each fold. This setup enables a more generalisable and comprehensive evaluation, as it avoids repeated testing on the same examples and better reflects performance under diverse data conditions.

For the HateMM dataset, the MF model was trained for 60 epochs with a batch size of 64. For the MultiHateClip (MHC) dataset, MF was trained for 20 epochs with a batch size of 32. For both datasets, the RAMF model was trained for 20 epochs using a batch size of 16.

Model training and testing were conducted on a laptop equipped with an Intel(R) Core(TM) i9-14900HX processor, 96 GB of system RAM, and an NVIDIA GeForce RTX 4090 Laptop GPU with 16 GB of VRAM. A fixed random seed of 2021 was used across all experiments, following the configuration reported in the original HateMM implementation. For CoT and AR experiments, use L40 46GB GPU memory experiments.

Importantly, some results reported in our experiments deviate from those in the original publications. This discrepancy primarily arises from our re-partitioning of the five-fold datasets, and this modification enables a more generalisable and comprehensive evaluation. In contrast, previous experimental work only fixed the data set division and changed the random seed five times (Lang et al., 2025; Wang et al., 2024a).

During inference, the language model was configured with a maximum of 2048 new tokens, temperature set to 0.7, top-p sampling with a threshold of 0.9, and sampling enabled. The pad token ID was set to the end-of-sequence token from the tokenizer. These settings were applied consistently across reasoning tasks, including CoT and AR, to ensure coherent yet diverse outputs.

## E  Efficiency Analysis

We compare RAMF with both a simple baseline (HateMM) (Das et al., 2023) and more advanced multimodal fusion models such as CMFusion (Zhang et al., 2024), as shown in Table 5. HateMM represents a lightweight MLP-based fusion paradigm, while CMFusion and RAMF adopt more sophisticated cross-modal interaction mechanisms. Compared to HateMM, RAMF introduces additional modules (LGCF and SCA), leading to moderate increases in parameters and FLOPs. However, when compared with advanced fusion models (e.g., CMFusion), RAMF achieves superior performance while maintaining comparable inference latency, demonstrating a favourable efficiency–effectiveness trade-off. Replacing SCA with MTA slightly reduces parameters but causes a 1.05% macro-F1 drop; substituting with standard attention increases parameters while yielding inferior performance. These results validate that SCA achieves an optimal efficiency-effectiveness balance. The VLM-based contrastive reasoning is performed as an offline preprocessing step.

Table 5: Efficiency and performance comparison in the HateMM dataset with Bert, MFCC and Vit. "Params" and "FLOPs" refer to the number of trainable parameters and the computational cost per forward pass, respectively. "Time" denotes the average per-sample inference latency. '→' means 'replaced by'.

| Model | Params | FLOPs | Time | MF1(%) |
|---|---|---|---|---|
| HateMM | 1.36M | 2.18G | 0.025ms | 79.3 |
| CMFusion | 0.71M | 3.65G | 0.21ms | 79.1 |
| RAMF | 3.78M | 3.10G | 0.20ms | 84.3 |
| SCA→MTA | 3.56M | 2.74G | 0.30ms | 83.25 |
| SCA→StdAttn | 4.61M | 2.80G | 0.17ms | 82.46 |

Table 6: End-to-end efficiency comparison including full pipeline cost on HateMM with Bert, MFCC and Vit. "Time" denotes the average per-sample full pipeline latency. "Tok/s" denotes VLM decoding throughput.

| Model | Time (s) | Tok/s | GPU Mem (GB) | MF1(%) |
|---|---|---|---|---|
| RAMF (Qwen3-VL-2B VLM) | 12.51 | 71.5 | 4.93 | 81.5 |
| RAMF (Qwen2.5-VL-7B VLM) | 26.55 | 14.5 | 10.26 | 81.2 |
| RAMF (Qwen2.5-VL-32B VLM) | 109.16 | 6.23 | 35.85 | 84.3 |

To complement the above analysis, we further report the end-to-end efficiency by explicitly including the VLM inference cost in the full pipeline, as shown in Table 6. Unlike Table 5, which focuses on the downstream multimodal fusion module and is directly comparable to prior work, Table 6 reflects the complete pipeline from raw video input to final prediction. All efficiency measurements are conducted on a single NVIDIA L40 GPU.

## F  Analysis of Reasoning Quality

Table 7: Analysis of reasoning quality across different VLMs on the HateMM dataset. "Len" denotes the average token length for each reasoning stage (TO: objective description, TH: hate-assumed, TN: non-hate-assumed), and "Avg Len" is the average across all three stages. "UQR" denotes the average unique word ratio. "Jaccard" measures lexical overlap between TH and TN, and "Cosine" denotes BERT-based semantic similarity between TH and TN.

| Model | TO Len | TH Len | TN Len | Avg Len | UQR | Jaccard | Cosine |
|---|---|---|---|---|---|---|---|
| Gemini-2.5-Flash | 448.2692 | 228.0546 | 305.3959 | 327.2399 | 0.5613 | 0.2158 | 0.9596 |
| GPT-5-mini | 735.4926 | 210.6407 | 244.5824 | 396.9052 | 0.5572 | 0.2200 | 0.9630 |
| Qwen2.5-VL-32B | 273.1132 | 125.2096 | 121.2690 | 173.1973 | 0.6652 | 0.1992 | 0.9662 |
| LLaMA-4-17B | 146.2322 | 105.2091 | 126.5967 | 126.0126 | 0.6188 | 0.2533 | 0.9694 |
| Qwen2.5-VL-7B | 44.6193 | 36.0599 | 50.6648 | 43.7814 | 0.9124 | 0.1487 | 0.9479 |

We further analyse the reasoning quality across different VLMs in terms of length and contrastiveness, as shown in Table 7. First, we observe that larger VLMs (e.g., Gemini and GPT) generate longer and more information-rich reasoning, as reflected by higher average token counts across all stages. In contrast, smaller models such as Qwen-7B produce significantly shorter outputs, indicating limited semantic coverage. Second, we evaluate the contrastiveness between hate-assumed (TH) and non-hate-assumed (TN) reasoning using both lexical overlap and BERT-based cosine similarity. Interestingly, while smaller models (e.g., Qwen-7B) exhibit lower similarity scores, this does not translate into better performance. Instead, we find that high-performing models tend to maintain moderate similarity while providing more structured and semantically grounded reasoning. These results suggest that effective contrastive reasoning requires not only divergence between TH and TN, but also sufficient semantic richness and consistency. Simply producing highly divergent reasoning without meaningful content may degrade downstream performance.

## G  Analysis of VLM Influence on MHC-Chinese Performance

Table 8: Effect of VLM choice on the Chinese subset of MHC under the same feature setting (mBERT + MFCC + ViViT).

| Reasoning VLM | Accuracy (%) | Macro-F1 (%) | F1 (H) (%) | Precision (H) (%) | Recall (H) (%) |
|---|---|---|---|---|---|
| Gemini 2.5 Flash | 71.22 ± 3.34 | 66.42 ± 4.29 | 54.39 ± 9.11 | 57.65 ± 8.13 | 54.70 ± 17.44 |
| GPT-5 mini | 71.33 ± 3.41 | 67.51 ± 4.17 | 57.04 ± 8.44 | 56.19 ± 4.04 | 59.76 ± 14.54 |
| Qwen-2.5-VL-32B | **72.37 ± 4.26** | **69.31 ± 2.39** | **60.22 ± 2.00** | **58.39 ± 6.09** | **63.75 ± 9.25** |

To gain a better understanding of why RAMF achieves a greater improvement in hate-related recall on the MHC Chinese subset, we conducted additional analysis by replacing the inference VLM whilst keeping the downstream feature settings unchanged, as shown in Table 8. Specifically, the recall rate with Qwen was 0.6375, whilst that with GPT-5 mini was 0.5976 and with Gemini 2.5 Flash was 0.5607. It is worth noting that even when using different VLMs, RAMF maintains a consistent performance improvement over the baseline model without a VLM (Table 1). This indicates that the performance gains primarily stem from the proposed reasoning-aware multimodal fusion architecture, rather than being dependent on a specific reasoning model. This performance can be attributed to the module design of LGCF and SCA, which are capable of capturing short-term salient cues and fine-grained cross-modal interactions.

At the same time, the differences between the various VLMs indicate that the choice of inference generator influences the magnitude of the performance gains. In particular, Qwen's superior performance demonstrates that better alignment with Chinese video contexts and higher-quality inference can further enhance the performance of hate-sensitive content detection, particularly in terms of recall.

Overall, these findings demonstrate that RAMF provides a robust and general-purpose reasoning-aware fusion framework, the effectiveness of which can be further enhanced through the use of more powerful and context-adaptive reasoning generators.

## H  Effectiveness of Contrastive Reasoning in a Unimodal Setting

Table 9: Unimodal text-only ablation study evaluating the impact of contrastive reasoning in HateMM dataset with Bert, MFCC and Vit. Acc: Accuracy; MF1: Macro-F1; P(H): Precision for hate class; R(H): Recall for hate class.

| Model | Acc (%) | MF1 (%) | P(H) (%) | R(H) (%) |
|---|---|---|---|---|
| Text only | $79.81 \pm 2.98$ | $78.54 \pm 2.85$ | $77.90 \pm 4.66$ | $70.08 \pm 7.28$ |
| Text + TO | $80.65 \pm 2.79$ | $79.71 \pm 2.48$ | $76.46 \pm 6.19$ | $75.94 \pm 7.19$ |
| Text + Full Reasoning | $82.31 \pm 3.13$ | $81.19 \pm 3.23$ | $80.58 \pm 4.95$ | $73.87 \pm 7.70$ |

To isolate the contribution of contrastive reasoning from multimodal fusion, we conduct a unimodal text-only ablation study. Specifically, we compare three settings: (1) using only the transcript (Text only), (2) augmenting the transcript with objective descriptions generated by the VLM (Text + TO), and (3) incorporating full contrastive reasoning, including objective, hate-assumed, and non-hate-assumed interpretations (Text + Full Reasoning). As shown in Table 9, introducing objective descriptions already improves performance over the text-only baseline, indicating that VLM-generated grounding provides useful semantic context. This demonstrates that the complementary semantic perspectives introduced by contrastive reasoning effectively enhance the model's ability to capture nuanced and context-dependent hateful intent, even in a unimodal setting. These results confirm that the performance improvements are not solely due to multimodal fusion, but also stem from the proposed reasoning mechanism itself.

## I  Comprehensive Failure Case Analysis

Figure 13 and 14 present representative failure cases of the proposed RAMF framework. For each case, we provide a modality-wise breakdown of visual, textual, and audio cues, together with an analysis explaining why the model fails under these specific conditions. The cases illustrate remaining challenges for reasoning-aware multimodal fusion, particularly in scenarios involving ambiguous contextual signals, conflicting modality cues, or nuanced pragmatic language use. This analysis complements the quantitative results by highlighting the limitations of the proposed approach.

| | Case 1 |
|---|---|
| **Ground True: Hate** |  |
| **Visual** | The video consists of a sequence of scenes from a video game environment, depicting characters gathering near a pickup truck, driving through urban and industrial areas, and engaging in combat-related actions such as running, aiming, and shooting. The visuals primarily convey action-oriented gameplay elements typical of crime or shooter-themed games. No explicit hateful symbols, gestures, or identity-targeting visual cues are present throughout the video. |
| **Text** | The transcribed text mainly describe narrative actions and situational dialogue related to movement, preparation, and confrontation (e.g., characters being ready to go, pointing weapons, or coordinating actions). The language does not contain explicit slurs or direct attacks against protected groups. However, certain phrases can implicitly allude to violence or aggression without clearly specifying an identity-based target, making the hateful intent highly context-dependent and difficult to infer from surface-level textual semantics alone. |
| **Audio** | The background audio features a lively and upbeat background music (BGM), creating a cheerful and energetic atmosphere that contrasts sharply with the violent actions depicted in the video. No explicit hateful speech or discriminatory verbal expressions are present in the audio channel. The positive emotional tone of the music acts as a strong neutralizing signal, potentially misleading the model into interpreting the content as entertainment-oriented rather than harmful. |
| **Analysis** | This case illustrates a failure scenario where the baseline model incorrectly classifies the video as non-hateful due to the absence of explicit hate cues in any single modality. The visual content appears as generic video game violence, the textual modality lacks direct identity-based attacks, and the audio modality introduces a strong positive emotional bias through cheerful background music. The multimodal inconsistency—violent actions paired with upbeat audio—dilutes the perceived severity of the content and obscures potential implicit hateful intent. Without deeper contextual reasoning to reconcile these conflicting signals, the baseline model overly relies on surface-level modality cues and fails to capture the nuanced, implicit nature of hate in this scenario. This highlights the limitation of standard multimodal fusion approaches when faced with subtle or masked hate expressions, especially in cases where audio or visual styles distract from underlying harmful semantics. |

Figure 13: Representative failure case 1.

| Case 2 | |
|---|---|
| **Ground True: Non hate** |  |
| **Visual** | The video shows a shirtless Black individual standing outdoors in what appears to be a parking lot, with cars and greenery visible in the background. The individual wears a beaded necklace and uses exaggerated facial expressions and playful gestures, including puckering lips and animated hand movements. Surrounding people are visible reacting with smiles and laughter. |
| **Text** | The transcript contains the phrase "you nigger", which is a highly offensive racial slur when interpreted in isolation. Due to its strong lexical association with hate speech, the presence of this term dominates the textual signal and triggers hate detection mechanisms. |
| **Audio** | The speaker's tone is energetic and playful, accompanied by laughter from people nearby. The laughter appears to be a natural response to the individual's exaggerated expressions and humorous behavior, rather than mockery or derision. No aggressive shouting, hostile intonation, or emotionally negative prosody is present. |
| **Analysis** | This case represents a false positive caused by over-reliance on surface-level textual cues without sufficient integration of cultural and contextual information across modalities. While the transcript includes a highly offensive racial slur, the visual and audio signals strongly indicate a playful, in-group interaction characterized by humor, exaggerated expression, and mutual amusement. The laughter and relaxed body language suggest social bonding rather than ridicule or hate. The baseline model fails to account for reclaimed or intra-group language usage and lacks pragmatic reasoning to reconcile the contradiction between explicit textual signals and non-hostile visual–audio context. This highlights a key limitation of current hate detection systems: lexical sensitivity without contextual grounding can lead to misclassification, particularly in cases involving culturally specific language use and expressive, non-adversarial social interactions. |

Figure 14: Representative failure case 2.

