# OpenReview forum: "Reasoning-Aware Multimodal Fusion for Hateful Video Detection"
_TMLR — Accepted by TMLR_

### Review · Reviewer_VJSH · 2026-03-12

**Summary Of Contributions:**

The paper proposes a systematic way to detect hateful videos. It introduces local-global context fusion (LGCF) and semantic cross attention mechanism to allow interaction between the modalities. To address the challenge of capturing nuanced hateful content the authors provide a structured 3-stage prompt that asks fro general objective description, for hate related inference and for non-hate references.
The paper shows interesting evaluation and ablation studies showing the improvement introduced by their method.

Strengths:

1.	Contributions of an Interesting and novel integration of techniques such as Das et. al., & Golovneva et. al.  to show improvement in performance on task challenges.

2.	Experimentation and comparison with various methods show the merits of the method

3.	Thorough ablation study

4.	Good treatment of related work

Weaknesses:

1.	Some parts require additional clarity and organization

2.	I would expect to see some error bars or STD around results to confirm that the improvement is significant and consistent, in particular since there is a 70/10/20 split. This needs to be for the results reported in the paper and not only for the one in the supplemental material

3. some of the mechanisms proposed could also be tested on different binary clasisifcation problems to reinforce the strength of the proposed mechanism in which the VLM is requested to provide support for the hypotheisis and also for the negative. But this is minor.

**Audience:**

Yes

**Audience Explanation:**

the problem is intesrting to TMLR audiance, and the algorithm proposed introduces improvement over SoTA

**Broader Impact Concerns:**

No such concerns

**Claims And Evidence:**

Yes

**Claims Explanation:**

Overall yes, there is a good support for the claims both in quantitative and qualitative results.  but as I;ve written there still needs to be a standard deviation reported with the results.

**Requested Changes:**

1.	Clarity: The Introduction doesn’t state explicitly what are Challenge 1 and Challenge 2, yest the authors address their solutions explicitly, please address these two challenges clearly
2.	In section 3.1. the content are not problem formulation but mostly problem solution, please state the problem clearly.
3.	Also, In section 3.1 it is not clear what is HateMM architecture and what is yours, your contribution in Figure 1 and in the text should be stated clearly
4.	There is switch of notation in section 3.4 compared to image 1: what is $X^m$ ? is that from the video encoder? It is unclear
5.	Where does $Z^m$ come from in equation (3)?
6.	What does the square brackets of W mean, if g is applied element wise to the local and global vectors?? Is it an entry? Or some kind of multiplication? What are the entries of W?
7.	Please what are the “Key Query” space structure.
8.	What is the y-axis in Fig 9
9.	Typo in claiming upper panel and lower panel in Figure 9, its Left and right panels
10.	Should I assume that the baseline in Figure 10 refers to HateMM?

---

> ### Author Response · Authors · 2026-04-02
> **Response to Reviewer VJSH (Part 1)**
>
> We sincerely thank the reviewer for the constructive comments and helpful suggestions. We have carefully revised the manuscript to improve clarity, rigor, and presentation. Below we respond to each comment and describe the corresponding revisions.
>
> ---
>
> ### **Response to Weakness 1**: Some parts require additional clarity and organization.
>
> To address this, the Introduction and Sections 3.1–3.6 have been reorganized and clarified.
>
> ---
> ### **Response to Weakness 2**: Missing error bars / standard deviation to assess statistical significance.
>
> To address this, we explicitly report standard deviations for all main results (Table 1) obtained from the 5-fold cross-validation setup.
>
> ---
>
> ### **Response to Weakness 3**: Limited validation on other binary classification problems.
>
> We agree that extending the evaluation to other binary classification tasks would be valuable. Given the scope of this work, we focus on hateful video detection. We will explore broader applications of the proposed framework in future work.

---

> > ### Author Response · Authors · 2026-04-02
> > **Response to Reviewer VJSH (Part 2)**
> >
> > Table 1: Video classification performance (%) (five-fold average). Standard deviations are also reported in percentage points. MF1: Macro-F1; Acc: Accuracy; P(H): Precision for hate class; R(H): Recall for hate class. The MF row corresponds to a configuration without VLM inference.
> >
> > | Model        | HateMM MF1 | HateMM Acc | HateMM P(H) | HateMM R(H) | MHC-C MF1 | MHC-C Acc | MHC-C P(H) | MHC-C R(H) | MHC-E MF1 | MHC-E Acc | MHC-E P(H) | MHC-E R(H) |
> > |:-------------|-----------:|-----------:|------------:|------------:|----------:|----------:|------------:|------------:|----------:|----------:|------------:|------------:|
> > | **UniModal** |            |            |             |             |           |           |             |             |           |           |             |             |
> > | BERT         | 78.6±2.0   | 79.5±1.4   | 74.3±2.3    | 74.9±6.9    | 58.8±4.8  | 63.1±4.0  | 41.8±6.2    | 49.2±14.5   | 62.6±1.1  | 65.1±1.4  | 49.4±2.3    | 58.6±7.1    |
> > | HXP          | 80.6±1.3   | 81.2±1.4   | 74.7±3.2    | 80.8±3.8    | -         | -         | -           | -           | 61.0±1.4  | 67.8±1.4  | 54.0±3.9    | 56.8±5.0    |
> > | MFCC         | 66.9±2.1   | 67.7±2.7   | 58.9±3.8    | 67.1±8.4    | 54.4±4.6  | 60.3±3.0  | 38.8±6.1    | 39.2±14.7   | 47.8±1.9  | 58.4±2.3  | 31.2±2.3    | 21.0±11.8   |
> > | CLAP         | 72.2±1.5   | 74.2±1.6   | 71.1±3.6    | 59.7±2.4    | 59.1±3.4  | 66.2±2.0  | 48.2±4.9    | 38.0±7.7    | 57.9±2.3  | 64.8±2.7  | 47.4±3.7    | 36.6±2.8    |
> > | ViT          | 71.2±2.6   | 72.8±2.6   | 67.1±3.5    | 62.6±5.2    | 61.9±3.9  | 65.5±4.1  | 48.2±5.7    | 51.3±9.2    | 56.6±5.2  | 61.2±3.5  | 43.8±6.8    | 41.9±11.6   |
> > | CLIP         | 73.4±2.8   | 74.6±3.1   | 69.5±4.9    | 66.4±2.7    | 61.2±2.1  | 65.6±3.0  | 48.1±5.0    | 48.8±3.7    | 63.6±2.3  | 67.3±3.2  | 52.2±4.8    | 52.7±2.9    |
> > | **LLM**      |            |            |             |             |           |           |             |             |           |           |             |             |
> > | GPT-4o       | 78.0±1.1   | 78.2±1.0   | 66.8±2.4    | 89.8±2.8    | 51.0±2.1  | 70.4±3.7  | 71.8±4.1    | 16.4±2.6    | 63.7±3.8  | 72.4±3.2  | 70.1±7.5    | 34.2±4.9    |
> > | Qwen-Max     | 66.8±2.4   | 67.0±2.4   | 55.2±2.9    | 92.3±2.9    | 61.6±1.6  | 68.4±2.3  | 53.5±7.8    | 40.3±3.2    | 70.2±4.8  | 73.0±4.3  | 60.6±7.7    | 62.3±6.8    |
> > | LLaMA3.1     | 72.1±3.1   | 72.7±2.5   | 61.5±4.7    | 73.9±4.1    | -         | -         | -           | -           | 69.1±3.5  | 74.3±2.8  | 67.2±8.2    | 49.1±5.6    |
> > | **VLM**      |            |            |             |             |           |           |             |             |           |           |             |             |
> > | Qwen-VL      | 70.3±1.8   | 70.3±1.8   | 58.2±2.9    | 90.8±2.9    | 60.8±2.9  | 70.3±3.6  | 59.8±7.5    | 32.4±4.5    | 71.3±3.1  | 75.7±2.7  | 69.4±3.6    | 53.3±7.2    |
> > | LLaMA4       | 69.8±3.3   | 69.8±3.2   | 57.8±4.1    | 91.2±1.6    | -         | -         | -           | -           | 69.4±2.7  | 74.4±2.3  | 66.9±3.9    | 50.2±7.5    |
> > | **Multimodal (T1A1V1)** | | | | | | | | | | | | |
> > | HateMM      | 79.3±2.0   | 79.8±2.2   | 73.9±2.4    | 79.5±5.6    | 61.0±2.7  | 68.3±2.3  | 51.8±3.3    | 53.1±11.6   | 62.3±5.1  | 67.6±5.0  | 54.0±10.6   | 41.0±5.8    |
> > | CMFusion    | 79.1±1.7   | 80.2±2.0   | 77.1±4.0    | 72.2±5.0    | 61.4±2.2  | 68.6±2.1  | 53.5±6.1    | 39.3±5.2    | 60.8±2.2  | 66.8±3.1  | 52.3±4.9    | 40.6±3.2    |
> > | MoRE        | 81.0±2.4   | 82.1±2.3   | 80.0±4.1    | 73.5±2.3    | 60.2±2.1  | 69.6±2.4  | 57.1±2.8    | 31.7±5.7    | 62.8±3.4  | 67.5±3.6  | 51.6±6.1    | 47.8±3.2    |
> > | MF          | 82.3±3.1   | 82.9±3.2   | 78.0±3.8    | 80.2±3.9    | 65.9±2.0  | 70.3±2.8  | 55.1±2.6    | 53.2±5.8    | 63.5±4.3  | 67.8±4.3  | 53.3±5.8    | 50.9±12.0   |
> > | RAMF        | 83.7±2.8   | 84.3±2.7   | 78.6±3.7    | 83.7±4.5    | 69.3±2.3  | 72.4±2.1  | 58.4±6.0    | 63.8±9.2    | 64.1±6.2  | 68.5±3.1  | 53.2±5.6    | 52.2±16.9   |
> > | **Multimodal (T2A2V2)** | | | | | | | | | | | | |
> > | HateMM      | 82.4±2.4   | 83.0±2.6   | 79.4±4.1    | 80.2±5.1    | 63.3±1.6  | 69.0±4.5  | 51.3±7.5    | 47.3±15.6   | 68.4±4.7  | 72.3±2.1  | 59.1±5.9    | 57.1±12.9   |
> > | CMFusion    | 81.9±2.1   | 82.8±2.0   | 80.2±2.6    | 75.8±4.1    | 60.6±3.8  | 66.7±3.2  | 49.0±4.7    | 42.4±7.7    | 67.7±2.7  | 71.7±2.2  | 60.6±6.6    | 55.1±1.0    |
> > | MoRE        | 82.1±2.1   | 82.9±2.4   | 77.0±3.4    | 79.7±5.6    | 62.5±4.5  | 69.1±4.0  | 54.1±7.1    | 41.2±12.1   | 67.4±3.5  | 72.5±4.1  | 61.1±8.5    | 49.3±13.1   |
> > | MF          | 83.2±3.3   | 83.7±3.5   | 78.3±6.0    | 82.6±2.9    | 66.2±4.4  | 70.3±4.1  | 55.2±8.1    | 51.9±8.4    | 69.6±4.3  | 73.4±3.3  | 62.1±7.8    | 56.8±9.0    |
> > | RAMF        | 85.1±1.9   | 85.6±1.1   | 79.8±1.9    | 85.5±5.4    | 70.9±4.3  | 74.5±4.0  | 61.3±6.4    | 62.2±11.5   | 71.7±4.3  | 74.0±4.1  | 62.1±9.3    | 67.4±11.0   |

---

> ### Author Response · Authors · 2026-04-02
> **Response to Reviewer VJSH (Part 3)**
>
> ### **Response to Requested Changes 1**: The Introduction does not explicitly state Challenge 1 and Challenge 2.
>
> We agree and have revised the Introduction to explicitly define the two core challenges:
>
> - **Challenge 1: Nuanced Context Understanding**. Hateful videos often convey harmful intent through nuanced contextual cues spanning time and modalities, such as the temporal resonance between specific visuals and statements, or the implicit linkage between visual focus and auditory content.
> - **Challenge 2: Multimodal Semantic Fusion**. Current methods follow a standard of extracting and fusing features from video frames, audio, and transcribed text, but lack effective multimodal semantic interaction.
>
> ---
>
> ### **Response to Requested Changes 2 & 3**: Section 3.1 is not a clear problem formulation, and it is unclear what belongs to the baseline (HateMM) versus the proposed method.
>
> To address this, we have restructured this part of the paper. Specifically, we separate the original Section 3.1 into two parts:
> - Section 3.1: Problem Formulation
> - Section 3.2: Framework Overview
> Section 3.1 has been rewritten as a formal problem formulation, and a new Section 3.2 has been introduced for model overview, with a clearer separation between baseline and proposed method.
> ---
>
> ### **Response to Requested Changes 4**: Notation $X^m$ is unclear.
>
> We clarified that $X^m$ denotes the embedding sequence produced by the encoder of modality $m$:
> $[X^m \in {R}^{L \times D_m}, \quad m \in {T, A, V}]$. Section 3.5 now explicitly defines $X^m$.
>
> ---
>
> ### **Response to Requested Changes 5**: Where does $Z^m$ come from?
>
> We clarified that $Z^m$ is the fused representation produced by the LGCF module:
> $[Z^m = g \odot v_{\text{local}} + (1-g) \odot v_{\text{global}}]$. The text now explicitly states that $Z^m$ is the final fused modality representation.
>
> ---
>
> ### **Response to Requested Changes 6**: What does $W$ represent?
>
> We clarify that the gating function is implemented as a standard linear transformation applied to the concatenation of local and global representations:
> $[
> g = \sigma\left(W \left( v_{\text{local}} \oplus v_{\text{global}} \right) + b \right)
> ]$
> where $v_{\text{local}}, v_{\text{global}} \in {R}^{D}$, $\oplus$ denotes vector concatenation, and $W \in {R}^{D \times 2D}$ is a learnable weight matrix. The concatenated vector lies in ${R}^{2D}$, and the output $g \in {R}^{D}$ is applied element-wise. We have corrected the notation and explicitly clarified the operation (concatenation + linear transformation) and dimensions in Section 3.5.
>
> ---
>
> ### **Response to Requested Changes 7**: What is the “Key Query” space structure?
>
> We clarify that this refers to the spatial structure of the query--key attention matrix. In our implementation, the attention tensor is treated as a structured representation, and convolution is applied over this structure to capture dependencies across attention heads and positions. The terminology has been revised to “spatial structure of the query--key attention matrix” in Section 3.6.
>
> ---
>
> ### **Response to Requested Changes 8**: What is the y-axis in Figure 9?
>
> We have now added a formal definition. The y-axis represents the normalized contribution score:
> $[
> c_i = \text{softmax}\left(\sum_{j=1}^{N} A_{j,i}\right)
> ]$ The definition is now explicitly included in both the main text and the figure caption of Figure 9.
>
> ---
>
> ### **Response to Requested Changes 9**: Typo in Figure 9 panels.
>
> We have corrected the inconsistency. The panels are now consistently referred to as left and right.
>
> ---
>
> ### **Response to Requested Changest 10**: Is the baseline in Figure 10 HateMM?
>
> We have explicitly clarified in the main text that the baseline in Figure 10 refers to the HateMM model.

---

> ### Comment · Reviewer_VJSH · 2026-04-20
> **Response to authors revision**
>
> I am happy with how the autorhs addressed my comments. I am recommending to accept.

---

### Review · Reviewer_AgF3 · 2026-03-19

**Summary Of Contributions:**

The paper proposes the RAMF framework (Reasoning-Aware Multimodal Fusion) aimed at classification of videos as Hateful or not. The authors describe the drawbacks of current approaches as (1) lack of nuanced context understanding and (2) lack of fine-grained semantic fusion to capture both local salient features and global temporal patterns in the data. To address these drawbacks, first the proposed framework uses a VLM component to generate three different descriptions of the video (frames + transcript) containing objective description, hate-assumed description, and non-hate assumed description. Then the proposed a Local-Global Context Fusion (LGCF) component with Semantic Cross-Attention (SCA) captures fine-grained semantics between the different modalities of data. Particularly, the SCA design inspired by Multi-token attention (Golovneva et. al. 2025) is a nice novel contribution in this paper especially in the context of fusing multiple different modalities of data available in videos.

Experiments are conducted on the HateMM and MultiHateClip datasets. Although the datasets are curated (as opposed to "in the wild"), the datasets contain real-world videos and although the number of videos in these datasets is small (~1000), it is quite standard in hateful video detection space and is well consistent to previous works and sets up for a valid comparison with previous works. Comprehensive quantitative and qualitative studies are reported and the proposed approach outperforms the previous works and baselines. Furthermore, the ablation study reports the contributions of each component of the proposed framework.

Overall, this is a well-rounded study for hateful video detection.

**Additional Comments:**

Replacing the Qwen 2.5-VL-32B model with Llama 4-17B results in a marginal drop in performance. In the interest of using this framework in production environments, It would be interesting to see how the pipeline performs with an even smaller VLM.

**Audience:**

Yes

**Audience Explanation:**

This paper proposed a novel framework outperforming the SotA approaches for hateful video detection. This work would be interesting for individuals working/interested in the content-moderation space. Also this work showcases the impact of the VLM component in content-moderation which is interesting.

**Broader Impact Concerns:**

The paper contains a sufficient broader impact statement.

**Claims And Evidence:**

Yes

**Claims Explanation:**

The proposed RAMF framework achieves State-of-the-Art (SotA) performance as shown in Table-1. The experiments are well designed with k-fold validation (k=5). The ablation study further proves the effectiveness of the LGCF and SCA modules. More specifically SCA is shown to be superior to different attention mechanisms including the Multi-token attention mechanism in the ablation study.

**Requested Changes:**

1. The authors label the generation of descriptions (objective, hate-assuming, non-hate-assuming) as "adversarial reasoning". Though I can try and wrap my head around the terminology, "adversarial" can be confusing to the readers in the AI space. Typically, "adversarial" implies to the min-max optimization problem like in Generative Adversarial Networks (GANs). However, in this work the generation of hate-assumed and non-hate assumed descriptions does not sound very adversarial. A better word would be "Contrastive".

2. In section D.1 (Efficiency analysis), line 2,  "...,inference latency rising from 0.025ms to 0.20ms" which is although a 700% increase, I wonder if this includes the VLM's "adversarial reasoning" step. Please clarify.

---

> ### Author Response · Authors · 2026-04-02
> **Response to Reviewer AgF3**
>
> We would like to express our sincere gratitude to the reviewers for their positive feedback and constructive comments. We are delighted that the reviewers found our study to be comprehensive and acknowledged the validity of the proposed mechanisms and experimental design. We are also grateful for the reviewers’ valuable suggestions, which have helped to further enhance the clarity and presentation of the manuscript.
>
> ### **Response to Requested Change 1: Terminology of “adversarial reasoning”**
>
> To improve clarity and avoid potential misunderstanding, we have revised the terminology throughout the manuscript. We have updated the terminology consistently throughout the paper.
>
> ---
>
> ### **Response to Requested Change 2: Latency clarification in Efficiency Analysis**
>
> We clarify that the VLM-based reasoning component is performed as an offline preprocessing step and is not included in the reported inference latency. The latency reported in the paper reflects only the runtime of the multimodal fusion model during deployment, making it directly comparable to prior multimodal approaches.
>
> In addition, we have revised the efficiency analysis to provide clearer context. Specifically, we explicitly compare RAMF with both simple MLP-based baseline (HateMM) and more advanced fusion models (CMFusion).
>
> Table 5: Efficiency and performance comparison on the HateMM dataset (BERT + MFCC + ViT). Params: number of trainable parameters. FLOPs: computational cost per forward pass. Time: average per-sample inference latency. → means "replaced by"
>
> | Model              | Params | FLOPs | Time   | MF1 (%) |
> |--------|-------:|------:|-------:|--------:|
> | HateMM             | 1.36M  | 2.18G | 0.025 ms | 79.3   |
> | **CMFusion**       | **0.71M** | **3.65G** | **0.21 ms** | **79.1** |
> | RAMF               | 3.78M  | 3.10G | 0.20 ms | 84.3   |
> | SCA → MTA          | 3.56M  | 2.74G | 0.30 ms | 83.25  |
> | SCA → StdAttn      | 4.61M  | 2.80G | 0.17 ms | 82.46  |
> ---
>
> ### **Response to Additional Comment: Performance with smaller VLMs**
>
> To address this, we have extended our analysis by including experiments with multiple VLMs of different scales. As shown in the table below (Table 3), we compare several models with varying capacities and observe that while smaller models lead to some performance degradation, the overall framework remains robust across different VLM choices.
>
> Table 3: Performance comparison of different vision-language models (VLMs) used in the RAMF framework on the HateMM dataset with BERT, MFCC, and ViT. All values are reported in percentage (%), with standard deviations also expressed in percentage.
>
> | VLM              | Acc (%)        | MF1 (%)        |
> |------------------|--------------:|--------------:|
> | Gemini-2.5-Flash | 84.17 ± 1.80  | 83.38 ± 1.54  |
> | GPT-5-mini       | 84.72 ± 3.46  | 83.96 ± 3.14  |
> | Qwen2.5-VL-32B   | 83.75 ± 2.81  | 84.26 ± 2.72  |
> | LLaMA-4-17B      | 83.62 ± 2.91  | 84.35 ± 2.86  |
> | Qwen2.5-VL-7B          | 82.69 ± 3.92  | 81.81 ± 3.86  |
> | Qwen3-VL-2B | 82.59 ± 2.87 | 81.49 ± 3.07 |

---

### Review · Reviewer_kz4Z · 2026-03-21

**Summary Of Contributions:**

This paper addresses two core challenges in hateful video detection by proposing the Reasoning-Aware Multimodal Fusion (RAMF) framework integrated with adversarial reasoning, the Local-Global Context Fusion (LGCF) module, and the Semantic Cross Attention (SCA) mechanism. The proposed method achieves state-of-the-art performance on two real-world datasets (HateMM and MultiHateClip), with significant improvements in Macro-F1 and hate class recall. The research is highly relevant to real-world content moderation needs, with innovative method design and a relatively rigorous experimental setup. However, the paper suffers from critical gaps in dataset details, model design justification, experimental analysis depth, robustness validation, and interpretability. Multiple ambiguous descriptions of key components and insufficient supplementary analyses limit the comprehensiveness and reproducibility of the research.

Major revisions are required before the paper can be considered for acceptance.

**Audience:**

Yes

**Audience Explanation:**

This paper addresses two core challenges in hateful video detection—ineffective multimodal semantic fusion and poor nuanced contextual understanding, which matches the scope of TMLR

**Claims And Evidence:**

Yes

**Claims Explanation:**

The proposed method achieves state-of-the-art (SOTA) performance on two real-world datasets (HateMM and MultiHateClip), with significant improvements in Macro-F1 and hate class recall. The research is highly relevant to real-world content moderation needs, with innovative method design and a relatively rigorous experimental setup.

**Requested Changes:**

1. The paper only mentions that MHC includes Chinese/English subsets from Bilibili/YouTube but provides no details on annotation consistency, hate type distribution, video duration distribution, or modality completeness (e.g., proportion of videos without audio/text). It also fails to analyze the reasons for the more significant improvement in hate class recall on the Chinese MHC subset by RAMF.

2. The paper uses Qwen 2.5-VL-32B as the reasoning VLM only for "hardware constraints" but provides no justification for why this model is chosen over other mainstream VLMs (e.g., LLAVA, CLIP-VL). It only replaces Qwen with LLaMA 4-17B in ablation studies and lacks comparative analysis of the impact of VLMs with different parameter sizes and training data on reasoning quality and overall model performance.

3. This paper should discuss the difference of objective descriptions generated by VLM with:

(a) Weakly Supervised Video Anomaly Detection with Anomaly-Connected Components and Intention Reasoning, CVPR 2026

4. This paper should discuss the difference and relation of semantic context learning with the following paper:

(a) Learning Local Semantic Signals and Inter-class Discrepancy for Weakly Supervised Video Anomaly Detection, TMM 2026
(b) Action-semantic consistent knowledge for weakly-supervised action localization, TMM 2024

---

> ### Author Response · Authors · 2026-04-02
> **Response to Reviewer kz4Z (Part 1)**
>
> We sincerely thank the reviewer for the detailed and constructive feedback. We appreciate the reviewer’s recognition of the novelty and effectiveness of our framework. We also acknowledge the concerns regarding dataset details, model justification, and analysis depth, and have made substantial revisions to improve clarity, completeness, and reproducibility.
>
> ### **Response to Requested Change 1: Dataset details and analysis of performance differences**
>
> To address this, we have added a dedicated analysis covering dataset properties and model behaviour. Specifically, we added two new sections in the appendix:
>
> - Dataset Details (Appendix A), which reports annotation consistency, class distribution, and modality completeness
> - Analysis of VLM Influence on MHC-Chinese Performance (Appendix G), which investigates the performance gain on the Chinese subset. We have added a dedicated analysis together with a new table (Table 7) to explicitly examine the impact of the reasoning VLM on the MHC-Chinese subset. By replacing the VLM while keeping the rest of the framework unchanged, we isolate its effect and show that RAMF consistently improves performance over both the non-VLM baseline and alternative VLM choices. This demonstrates that the performance gains are primarily attributed to the proposed architecture, while the reasoning VLM further influences the magnitude of improvement, particularly in hate-class recall due to language alignment and reasoning quality.
>
> Table 7: Effect of VLM choice on the Chinese subset of MHC under the same feature setting (mBERT + MFCC + ViViT).
>
> | Reasoning VLM | Accuracy (%) | Macro-F1 (%) | F1 (H) (%) | Precision (H) (%) | Recall (H) (%) |
> |---|---:|---:|---:|---:|---:|
> | Gemini 2.5 Flash | 71.22 ± 3.34 | 66.42 ± 4.29 | 54.39 ± 9.11 | 57.65 ± 8.13 | 54.70 ± 17.44 |
> | GPT-5 mini | 71.33 ± 3.41 | 67.51 ± 4.17 | 57.04 ± 8.44 | 56.19 ± 4.04 | 59.76 ± 14.54 |
> | Qwen-2.5-VL-32B | 72.37 ± 4.26 | 69.31 ± 2.39 | 60.22 ± 2.00 | 58.39 ± 6.09 | 63.75 ± 9.25 |
>
> ---
>
> ### **Response to Requested Change 2: Justification and analysis of VLM choice**
>
> To address this, we have extended our experiments to include multiple VLMs of different scales and architectures. In the main text, we introduce additional comparisons (Table 3) to evaluate performance across different VLM choices, as shown below.
>
> Table 3: Performance comparison of different vision-language models (VLMs) used in the RAMF framework on the HateMM dataset with BERT, MFCC, and ViT. All values are reported in percentage (%).
>
> | VLM | Acc | MF1 |
> |---|---:|---:|
> | Gemini-2.5-Flash | 84.17 ± 1.80 | 83.38 ± 1.54 |
> | GPT-5-mini | 84.72 ± 3.46 | 83.96 ± 3.14 |
> | Qwen2.5-VL-32B | 83.75 ± 2.81 | 84.26 ± 2.72 |
> | LLaMA-4-17B | 83.62 ± 2.91 | 84.35 ± 2.86 |
> | Qwen2.5-VL-7B | 82.69 ± 3.92 | 81.81 ± 3.86 |
> | Qwen3-VL-2B | 82.59 ± 2.87 | 81.49 ± 3.07 |
>
> Furthermore, we provide a detailed analysis of reasoning quality across VLMs in the appendix (Appendix F). Specifically, we added a new appendix section with Table 6:
>
> - Analysis of Reasoning Quality (Appendix F), including detailed metrics such as reasoning length, unique word ratio, Jaccard similarity, and BERT-based cosine similarity.
>
> Table6: Analysis of reasoning quality across different VLMs on HateMM dataset. Len: average token length for each reasoning stage (TO: objective description, TH: hate-assumed, TN: non-hate-assumed). Avg Len: average across all three stages. UQR: average unique word ratio. Jaccard: lexical overlap between TH and TN. Cosine: BERT-based semantic similarity between TH and TN
>
> | Model             | TO Len  | TH Len  | TN Len  | Avg Len | UQR    | Jaccard | Cosine |
> |--------|--------:|--------:|--------:|--------:|--------:|--------:|--------:|
> | Gemini-2.5-Flash   | 448.2692 | 228.0546 | 305.3959 | 327.2399 | 0.5613 | 0.2158 | 0.9596 |
> | GPT-5-mini         | 735.4926 | 210.6407 | 244.5824 | 396.9052 | 0.5572 | 0.2200 | 0.9630 |
> | Qwen2.5-VL-32B           | 273.1132 | 125.2096 | 121.2690 | 173.1973 | 0.6652 | 0.1992 | 0.9662 |
> | LLaMA-4-17B        | 146.2322 | 105.2091 | 126.5967 | 126.0126 | 0.6188 | 0.2533 | 0.9694 |
> | Qwen2.5-VL-7B            | 44.6193  | 36.0599  | 50.6648  | 43.7814  | 0.9124 | 0.1487 | 0.9479 |
>
> These results provide deeper insight into how different VLMs influence reasoning quality and downstream performance.

---

> ### Author Response · Authors · 2026-04-02
> **Response to Reviewer kz4Z (Part 2)**
>
> ### **Response to Requested Changes 3 & 4: Relation to prior works on anomaly detection and semantic context learning**
>
> To address this, we have expanded the related work section to include a more detailed comparison with the suggested works. In particular:
>
> - We clarify the differences between our VLM-based reasoning approach and prior works on reasoning
> - We discuss the relation between our semantic fusion design and existing approaches to semantic context learning

---

### Review · Reviewer_zC7N · 2026-04-07

**Summary Of Contributions:**

The authors propose a modal-fusion hate-video detection framework. Specifically, they design Local-Global Context Fusion (LGCF) to capture local and global features and propose contrastive reasoning to capture nuanced hateful intent effectively. The framework consists of two steps; First, VLM contrastive reasoning is that VLM generate descriptions of videos with three different instructions. Second, LGCF module classifies multi-modal videos with descriptions generated in the first step using Semantic Cross Attention (SCA) mechanism. This framework shows high accuracy compared to other multi-modal methods for hate-video detection.

**Audience:**

Yes

**Audience Explanation:**

Hateful video detection is an important topic of trustworthy ai, and the use of VLM-generated reasoning for multimodal classification would interest researchers.

**Broader Impact Concerns:**

The broader impact statement adequately addresses the potential risks and limitations.

**Claims And Evidence:**

Yes

**Claims Explanation:**

The motivation and problem definition are clear and well-written. The proposed method is reasonable and shows high performance compared to existing methods.

**Requested Changes:**

Overall, the study is well motivated and the proposed method is reasonably designed. However, it would be better to address the following concerns.

1. Effectiveness of contrastive reasoning. The idea of generating contrastive outputs for safety alignment by utilizing opposing instructions has been explored in previous work [1]. How does the proposed contrastive reasoning compare to or differ from this approach? Could incorporating such contrastive decoding further improve the framework's performance? Furthermore, to demonstrate the effectiveness of contrastive reasoning, it would be valuable to show that contrastive reasoning improves performance over simple text-based detection in a unimodal setting.

[1] ROSE Doesn't Do That: Boosting the Safety of Instruction-Tuned Large Language Models with Reverse Prompt Contrastive Decoding

2. Comparison of efficiency. Since the framework relies on VLM-generated descriptions, the overall pipeline is likely to be computationally heavy. A comparison of end-to-end efficiency with other methods, including the VLM inference cost, would strengthen the paper. Additionally, does the framework still perform well when using a smaller VLM such as Qwen3-VL-2B?

---

> ### Author Response · Authors · 2026-04-10
> **Response to Reviewer zC7N**
>
> We sincerely thank the reviewer for the detailed and constructive feedback. We appreciate the recognition of our framework and the insightful suggestions. We have carefully revised the manuscript to improve clarity, analysis, and completeness.
>
> ---
>
> ### **Response to Requested Change 1:**
>
> #### **1) Effectiveness of contrastive reasoning and comparison with prior work**
>
> To address this, we have expanded the related work section 2.3 to include a more detailed comparison with the suggested work.
>
> While both approaches leverage opposing instructions, they differ fundamentally:
> - Prior work operates at the distribution level, adjusting token probabilities using positive and negative prompts. It produces a single output without explicitly modeling alternative reasoning.
> - In contrast, our proposed contrastive reasoning operates at the process level by explicitly generating and contrasting alternative reasoning trajectories (e.g., hate-assumed vs. non-hate interpretations), enabling reliable and interpretable semantic understanding
>
>
> ---
>
> #### **2) Potential combination with contrastive decoding**
>
> While our contrastive reasoning operates at the semantic representation level rather than decoding, we agree that
> integrating such contrastive decoding strategies could be a promising direction.
> In particular, combining inference-time contrastive decoding with our structured reasoning framework may
> further enhance robustness and performance. We will explore this direction in future work.
>
>
> ---
>
> #### **3) Unimodal validation of contrastive reasoning**
>
> We have added a unimodal text-only ablation study in Appendix H.
>
> Table 9: Unimodal text-only ablation study evaluating the impact of contrastive reasoning in HateMM dataset with Bert, MFCC and Vit. Acc: Accuracy; MF1: Macro-F1; P(H): Precision for hate class; R(H): Recall for hate class.
>
> | Model | Acc (%) | MF1 (%) | P(H) (%) | R(H) (%) |
> |------|--------|--------|----------|----------|
> | Text only | 79.81 ± 2.98 | 78.54 ± 2.85 | 77.90 ± 4.66 | 70.08 ± 7.28 |
> | Text + TO | 80.65 ± 2.79 | 79.71 ± 2.48 | 76.46 ± 6.19 | 75.94 ± 7.19 |
> | Text + Full Reasoning | **82.31 ± 3.13** | **81.19 ± 3.23** | **80.58 ± 4.95** | **73.87 ± 7.70** |
>
> This demonstrates that the proposed reasoning mechanism is effective even without multimodal fusion.
>
> ---
>
> ### **Response to Requested Change 2:**
>
> #### **1) End-to-end efficiency analysis**
>
> We have added an end-to-end efficiency analysis (Table 6) including VLM inference cost in Appendix E.
>
> Table 6: End-to-end efficiency comparison including full pipeline cost on HateMM with Bert, MFCC and Vit.
> "Time" denotes the average per-sample full pipeline latency. "Tok/s" denotes VLM decoding throughput.
>
> | Model | Time (s) | Tok/s | GPU Mem (GB) | MF1 (%) |
> |------|---------|-------|--------------|---------|
> | RAMF (Qwen3-VL-2B) | 12.51 | 71.5 | 4.93 | 81.5 |
> | RAMF (Qwen2.5-VL-7B) | 26.55 | 14.5 | 10.26 | 81.2 |
> | RAMF (Qwen2.5-VL-32B) | 109.16 | 6.23 | 35.85 | 84.3 |
>
> ---
>
> #### **2) Performance with smaller VLMs**
>
> We further evaluated the framework with Qwen3-VL-2B.
>
> Table 3:Performance comparison of different vision-language models (VLMs) used in the RAMF framework on the HateMM dataset with Bert, MFCC and Vit. All values are reported in percentage (\%), with standard deviations also expressed in percentage.
>
> | VLM | Acc | MF1 |
> |-----|-----|-----|
> | Gemini-2.5-Flash | 84.17 ± 1.80 | 83.38 ± 1.54 |
> | GPT-5-mini | 84.72 ± 3.46 | 83.96 ± 3.14 |
> | Qwen2.5-VL-32B | 83.75 ± 2.81 | 84.26 ± 2.72 |
> | LLaMA-4-17B | 83.62 ± 2.91 | 84.35 ± 2.86 |
> | Qwen2.5-VL-7B | 82.69 ± 3.92 | 81.81 ± 3.86 |
> | Qwen3-VL-2B | 82.59 ± 2.87 | 81.49 ± 3.07 |

---

### Decision · Action_Editor_Tbqo · 2026-05-15

**Recommendation:** Accept as is

**Audience:**

Yes

**Audience Explanation:**

obviously interest to ML community.

**Claims And Evidence:**

Yes

**Claims Explanation:**

The manuscript has been reviewed and agreed by the reviewers that it is clearly reliable.